# Partition-Then-Adapt: Combating Prediction Bias for Reliable Multi-Modal Test-Time Adaptation

**Guowei Wang**[1]    **Fan Lyu**[2]    **Changxing Ding**[1]*

[1]South China University of Technology

[2]National Laboratory of Pattern Recognition, Institute of Automation,
Chinese Academy of Sciences

eegw.wang@mail.scut.edu.cn, fan.lyu@cripac.ia.ac.cn, chxding@scut.edu.cn

## Abstract

Existing test-time adaptation (TTA) methods primarily focus on scenarios involving domain shifts in a single modality. However, they often prove ineffective when multiple modalities simultaneously undergo domain shifts, as they struggle to identify and utilize reliable samples within testing batches amid severe prediction bias. To address this problem, we propose **P**artition-**T**hen-**A**dapt (**PTA**), a novel approach combating prediction bias for TTA with multi-modal domain shifts. PTA comprises two key components: Partition and Debiased Reweighting (PDR) and multi-modal Attention-Guided Alignment (AGA). Specifically, PDR evaluates each sample's predicted label frequency relative to the batch average, partitioning the batch into potential reliable and unreliable subsets. It then reweights each sample by jointly assessing its bias and confidence levels through a quantile-based approach. By applying weighted entropy loss, PTA simultaneously promotes learning from reliable subsets and discourages reliance on unreliable ones. Moreover, AGA regularizes PDR to focus on semantically meaningful multi-modal cues. Extensive experiments validate the effectiveness of PTA, surpassing state-of-the-art method by 6.1% on Kinetics50-MC and 5.8% on VGGSound-MC, respectively. Code of this paper is available at https://github.com/MPI-Lab/PTA.

## 1 Introduction

Pre-trained models tend to encounter performance degradation when deployed due to source-target domain shifts. To mitigate this, TTA updates model parameters during inference using online data, making it essential for applications like visual understanding [31, 30] and embodied intelligence [6, 18]. Recent studies have expanded TTA to multi-modal scenarios (MM-TTA), such as multi-modal action recognition and event classification [43, 50, 9]. However, they primarily focus on single-modal domain shifts in multi-modal tasks, where only one modality undergoes a shift while others remain unaffected. In the real world, multi-modal domain shifts are more commonly encountered. For instance, variations in lighting and background noise can degrade the performance of cameras and acoustic sensors used in autonomous driving systems [43, 40]. *In this paper, we investigate the challenging problem of simultaneous domain shifts across multiple modalities in MM-TTA scenarios.*

Existing MM-TTA methods typically address single-modal domain shifts in multi-modal tasks, but they fall short in handling multi-modal domain shifts. This is mainly due to difficulties in identifying and effectively leveraging reliable samples. From Fig. 1 (a), we observe that as more modalities become contaminated, the pre-trained model gradually loses reliability in multi-modal fusion, causing fused representations of different classes to overlap. Consequently, as shown in Fig. 1 (b), this results in biased predictions. In the context of TTA, updating the pre-trained model with biased predictions

---

*Corresponding author.

39th Conference on Neural Information Processing Systems (NeurIPS 2025).

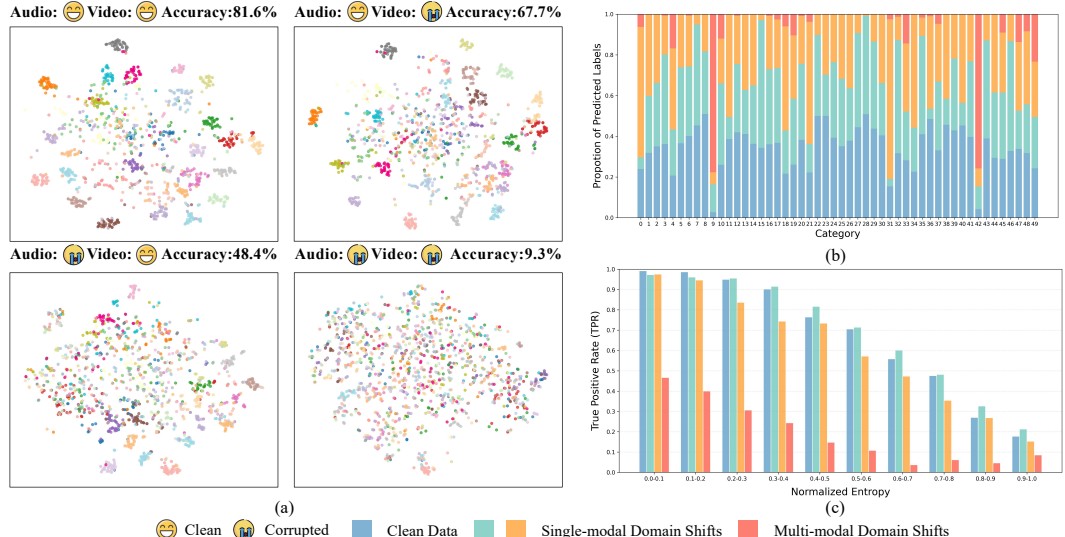

Figure 1: (a) t-SNE [35] visualizations of fused representations under clean, single-modal, and multi-modal domain shifts. (b) Proportions of predicted labels under the same domain shift conditions as (a). (c) Comparison of true positive rates for test data grouped by normalized entropy bins (interval: 0.1) . All experiments are conducted on the Kinetics50-C(MC) [43] with the pre-trained CAV-MAE [43] model. The audio and video noise types are "Crowd" and "Gaussian Noise", respectively.

rapidly accumulates errors, leading to collapsed trivial solutions [26], *i.e.*, limiting predictions to a narrow subset of classes. One popular way to mitigate this is to perform model adaptation using only high-confidence samples [43, 50, 9]. However, this strategy frequently struggles to adapt to multi-modal domain shifts. As illustrated in Fig. 1 (c), we observe a notable drop in the true positive rate for high-confidence samples, indicating that many of the selected "reliable" samples are actually *false positives*. In this case, updating the pre-trained model with these "reliable" samples amplifies the risk of error accumulation. Therefore, relying exclusively on prediction confidence fails to provide a reliable basis for sample selection and reweighting during TTA under multi-modal domain shifts.

To address the above challenges, we propose a Partition-Then-Adapt (PTA) method, which discovers and leverages reliable samples for TTA in the presence of multi-modal domain shifts. PTA consists of two key components, *i.e.*, Partition and Debiased Reweighting (PDR) and multi-modal Attention-Guided Alignment (AGA). Specifically, given test samples under multi-modal domain shifts, PTA first quantifies the prediction bias level for each sample. This bias is calculated based on how frequently the sample's predicted label appears relative to the average frequency of predicted labels within the batch. Samples with lower prediction bias are identified as potentially reliable, while those with higher bias are deemed potentially unreliable. It then employs a quantile-based reweighting scheme in the multi-modal adaptation that jointly assesses prediction bias and confidence levels, assigning positive and negative weighting factors for reliable and unreliable samples. By separately applying weighted entropy minimization and maximization to each group, it simultaneously promotes learning from reliable samples and discourages reliance on unreliable ones. Since the entropy maximization for unreliable samples may drive the attention map toward an undesired uniform distribution, potentially causing the model to attend to non-discriminative or noisy signals, we further introduce a multi-modal attention-guided alignment (AGA) method to mitigate this risk. Specifically, AGA uses the attention map distributions of reliable samples to guide the update of those from unreliable samples through a maximum mean discrepancy-based regularization. It ensures the attention maps distribution of unreliable samples remains aligned with the semantically meaningful cues from reliable samples, rather than being misled by undesired uniform distributions. The contributions of this work can be summarized as follows:

(1) We show that solely believing in prediction confidence may induce error accumulation under multi-modal domain shifts in TTA.

(2) We propose a novel PTA method, which consists of Partition and Debiased Reweighting (PDR) and multi-modal Attention-Guided Alignment (AGA). PDR offers a reliable sample selection and reweighting scheme that combats prediction bias for MM-TTA. AGA further regularizes PDR to focus on semantically meaningful multi-modal cues.

(3) PTA consistently outperforms all state-of-the-art methods on multi-modal tasks, with pronounced advantages under multi-modal domain shifts.

## 2 Related Work

**Test-Time Adaptation.** TTA [37, 19] aims to adapt a pre-trained model to the target domain using online data without ground-truth labels. Traditional works typically address four core challenges of TTA, *i.e.*, noisy pseudo-labels [3, 24], biased entropy [25, 16, 13, 38, 41], catastrophic forgetting [40], and miscalibrated batch normalization statistics [45]. To mitigate noisy pseudo-labels and biased entropy, existing methods employ sample selection/reweighting [25, 26, 16], pseudo-label refinement [3], and robust prototypes learning [5, 48, 39]. For catastrophic forgetting, memory banks [40], Fisher regularization [25, 33], and teacher-student frameworks [40, 5, 22, 34, 28] are commonly adopted to achieve prediction consistency. Meanwhile, miscalibrated statistics are typically corrected via source-target normalization statistics mixup [45, 7, 49]. Although effective in single-modal settings, their direct application to multi-modal scenarios often yields suboptimal results, as real-world multi-modal domain shifts involve more complex noise patterns [9].

**Multi-Modal Test-Time Adaptation.** Different from traditional TTA studies, which primarily handle single-modal data, MM-TTA addresses the challenge of domain shifts in multi-modal data. Pioneer works [29, 2, 1] attribute the challenges to intra- and cross-modal error accumulation, and conduct similar operations with traditional single-modal TTA methods, such as pseudo-labeling [37, 20] and teacher-student guidance [40, 5]. A recent work [43] advocates that the key of MM-TTA is mitigating increasing information discrepancies, *i.e.*, reliability bias, and proposes reliable fusion and robust adaptation as a solution. Its followers introduce attention bootstrapping [50], mutual information sharing [9] to further boost the performance. However, they focus on addressing single-modal domain shifts, where only one modality undergoes domain shift while others remain intact. In this paper, we demonstrate the robustness of our method on both single-modal and more practical and challenging multi-modal domain shifts.

## 3 Method

### 3.1 Problem Formulation

We define the problem using the example of **video** and **audio** co-classification for clarity. Consider a multi-modal pre-trained model $\mathcal{M}_\Theta = (\phi_v, \phi_a, \mathcal{F})$, where $\phi_v$ and $\phi_a$ are the encoders for video modality $v$ and audio modality $a$, respectively, and the block $\mathcal{F}$ is the fusion component with a prediction head. Let the output logits be $\mathbf{p}(\mathcal{X}) = \mathcal{M}_\Theta(\mathcal{X})$ for online mini-batch $\mathcal{X}$, where $\mathcal{X} = \{x_i\}_{i=1}^n = \{(x_1^v, x_1^a), (x_2^v, x_2^a), \cdots, (x_n^v, x_n^a)\}$. For sample $x \in \mathcal{X}$, the prediction can be computed by $\hat{y}(x) = \arg\max(\mathrm{softmax}(\mathbf{p}(x)))$. During MM-TTA, the model updates the tunable parameters $\tilde{\Theta} \subseteq \Theta$ with self-supervised loss functions, *e.g.*, entropy loss.

Previous MM-TTA methods [43, 50] focus on single-modal domain shifts, where only one modality is corrupted, which may deviate from real-world scenarios. In this paper, we focus on multi-modal domain shifts in MM-TTA, and propose a novel Partition-Then-Adapt (PTA) method. As shown in Fig. 2, PTA consists of Partition and Debiased Reweighting (PDR) and multi-modal Attention-Guided Alignment (AGA). PDR identifies reliable and unreliable candidates and reweights their contributions, and AGA regularizes PDR to focus on semantically meaningful multi-modal cues.

### 3.2 Partition and Debiased Reweighting

Sample reweighting is a widely adopted strategy in traditional TTA, designed to balance the influence of reliable and unreliable samples. Building on this approach, previous MM-TTA methods select and reweight samples simply guided by low entropy [25, 43, 9] or high softmax confidence [50]. They perform well when only one modality undergoes domain shift among multiple modalities, allowing the pre-trained model to leverage reliable, unaffected modalities to preserve prior knowledge [43, 9], thereby maintaining the reliability of selected samples.

However, as we demonstrate in the experimentation section (Tables 1-3 and 6-7), the effectiveness of existing reweighting strategies becomes less effective under multi-modal domain shifts compared to single-modal domain shifts. In such scenarios, corrupted modalities compromise the reliability of

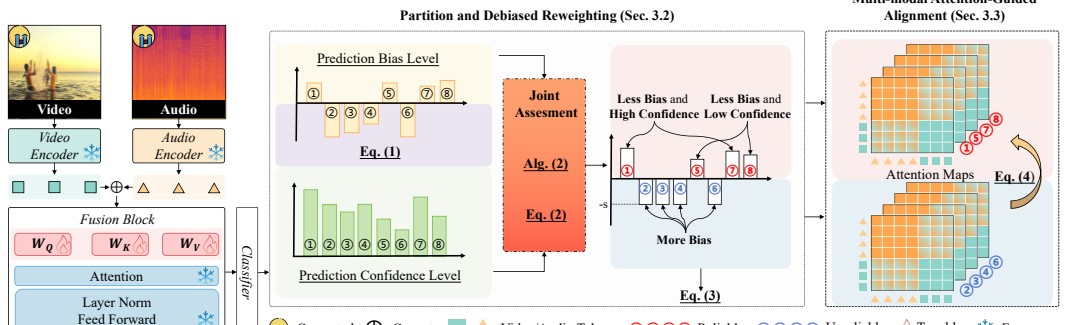

Figure 2: Overview of PTA. PTA first partitions the online data into two subsets, and jointly evaluates sample importance considering their prediction bias and confidence levels. It then adapts the pretrained model by weighted entropy minimization and multi-modal attention-guided alignment.

prior knowledge, degrade fused representations, and constrain predictions to a narrow set of classes. Unlike single-modal settings, noise in a single modality can propagate through the fusion process, disrupting the entire multi-modal system [9]. As more modalities are affected, the collapse of prior knowledge leads to a sharp decline in the true positive rate of confident predictions and a more pronounced increase in false positives. Consequently, confidence or entropy becomes an unreliable indicator for identifying and reweighting samples under multi-modal domain shifts. In such cases, adapting the model further increase the risk of prediction bias. *To address this challenge, we design a Partion and Debiased Reweighting (PDR) strategy that effectively combats prediction bias under multi-modal domain shifts.*

PTA first identifies reliable and unreliable samples, and respectively promote and restrain their contributions for better adaptation. Given a batch of multi-modal samples $\mathcal{X}$, with corresponding predicted categories $\hat{\mathcal{Y}}$. For $x \in \mathcal{X}$, the frequency of its predicted label can be expressed as $Z(x) = \mathbb{E}_{\hat{y}_i \in \hat{y}} \mathbf{1}(\hat{y}_i = \hat{y})$. Here, $\mathbf{1}(\cdot)$ is the indicator function that equals 1 when the condition is true. For samples in the mini-batch $\mathcal{X}$, we define their bias level of prediction as:

$$\mathcal{Z}(\mathcal{X}) = \bar{Z} - Z(\mathcal{X}) = \left\{ \bar{Z} - Z(x_1), \bar{Z} - Z(x_2), \ldots, \bar{Z} - Z(x_n) \right\}, \tag{1}$$

where $\bar{Z} = \mathbb{E}_{x \in \mathcal{X}} Z(x)$ is the mean frequency of the predicted labels, therefore representing the average bias level of all samples in the batch. Let the prediction confidence of $\mathcal{X}$ be denoted as $\mathcal{K}(\mathcal{X}) = \max(\text{softmax}(\mathbf{p}(\mathcal{X})))$. Based on Eq. (1), we partition $\mathcal{X}$ into two subsets: $\mathcal{X}^+$ for less biased predictions ($\mathcal{Z} \geq 0$) and $\mathcal{X}^-$ for more biased predictions ($\mathcal{Z} < 0$), yielding corresponding pairs of bias levels and confidences: $(\mathcal{Z}^+, \mathcal{K}^+)$ and $(\mathcal{Z}^-, \mathcal{K}^-)$. Building upon this partitioning, we introduce a novel approach that leverages the sample-wise bias levels for achieving more reliable reweighting adaptation.

To identify the reliability of test samples, we reweight those in $\mathcal{X}^+$ using their corresponding bias and confidence levels, $\mathcal{Z}^+$ and $\mathcal{K}^+$, via a quantile-based importance weighting scheme: $Q(\mathcal{Z}^+)$ and $Q(\mathcal{K}^+)$, where $Q(\cdot)$ denotes the quantile ranking operation inspired by [15, 32]. Specifically, we first sort all elements in $\mathcal{Z}$ and $\mathcal{K}$, then compute each element's quantile by averaging the positions of its first and last occurrences in the sorted sequence, normalized by the total number of elements. The details can be found in Algorithm 1. This quantile-based operation on $\mathcal{Z}^+$ and $\mathcal{K}^+$ serves to normalize sample influence and suppress outliers by mapping raw scores to a relative scale. By jointly ranking bias and confidence, it highlights samples that are both confident and less biased, enabling more reliable and robust adaptation. For a test sample $x \in \mathcal{X}$, the reweighting indicator function can be formulated as,

$$\mathbf{I}(x) = \begin{cases} Q(\mathcal{Z}^+(x)) \cdot Q(\mathcal{K}^+(x)), & \text{if } x \in \mathcal{X}^+, \\ -s, & \text{if } x \in \mathcal{X}^-. \end{cases} \tag{2}$$

The reliable samples in $\mathcal{X}^+$ are those with less prediction bias and high confidence, and the unreliable ones in $\mathcal{X}^-$ are those with large prediction bias. We set the weight of $\mathcal{X}^-$ to $-s$ ($s >= 0$), to reject samples with obviously biased predictions during entropy minimization. Therefore, the weighted entropy loss can be formulated as,

$$\mathcal{H}(\mathcal{X}) = -\mathbf{I}(\mathcal{X}) \cdot \sum \mathbf{p}(\mathcal{X}) \log \mathbf{p}(\mathcal{X}). \tag{3}$$

| **Algorithm 1** Quantile Ranking | **Algorithm 2** PTA |
|---|---|
| 1: **Input:** Sequence $\mathcal{T}$ | 1: **Input:** Trained model $\mathcal{M}_{\Theta}$, Data batch $\mathcal{X}$ |
| 2: **Output:** Quantile sequence $Q$ | 2: Calculate the prediction logits: $\mathbf{p}(\mathcal{X}) = \mathcal{M}_{\Theta}(\mathcal{X})$ |
| 3: $\mathcal{T}_{\text{sorted}} \leftarrow \text{sort}(\mathcal{T})$, $Q \leftarrow [\,]$ | 3: Reweighting each element in $\mathcal{X}$ using Eq. (1), Eq. (2), and Algorithm 1 |
| 4: **for each** $t \in \mathcal{T}$ **do** | 4: Calculate the weighted entropy loss using Eq. (3) |
| 5: $\quad$ Find first index $j$ where $\mathcal{T}_{\text{sorted}}[j] = t$ | 5: Calculate the MMD-based loss via Eq. (4) |
| 6: $\quad$ Find last index $k$ where $\mathcal{T}_{\text{sorted}}[k] = t$ | 6: Calculate the overall loss using Eq. (5) |
| 7: $\quad$ Compute $Q_t \leftarrow \frac{j+k}{2|\mathcal{T}|}$ | 7: Update the tunable parameters $\tilde{\Theta}$ of $\mathcal{M}_{\theta}$ |
| 8: $\quad$ Append $Q_t$ to $Q$ | |
| 9: **end for** | |
| 10: **Return** $Q$ | |

In summary, PTA takes both bias and confidence levels into account. Consequently, in the presence of numerous false positives, the high bias levels help reduce the corresponding weights, thereby mitigating error propagation during MM-TTA.

## 3.3 Multi-modal Attention-Guided Alignment

In Eq. (3), we introduce an entropy-based objective that assigns positive weights to reliable samples $\mathcal{X}^+$ and negative weights to unreliable ones $\mathcal{X}^-$, aiming to minimize the entropy of $\mathcal{X}^+$ while maximizing that of $\mathcal{X}^-$. $\mathcal{X}^+$ has robust prior knowledge compared to that of $\mathcal{X}^-$, and we will verify in Section 4.3. This mechanism, however, introduces a potential risks. In multi-modal learning, attention-based fusion [36, 23], especially self- and cross-modal attention, is widely used to integrate information from different modalities. Under the entropy maximization objective, the attention map $\mathcal{A}^-$ associated with $\mathcal{X}^-$ may converge toward a uniform distribution during adaptation. This can cause the model to attend to non-discriminative or noisy signals, which in turn interferes with the reliability and effectiveness of $\mathcal{A}^+$ in the fusion process. To mitigate this issue, based on the partition above, we propose a multi-modal Attention-Guided Alignment (AGA), which ensures that attention remains aligned with semantically meaningful multi-modal cues. We explicitly enforces the model to learn semantically meaningful multi-modal cues and push it as a guide for model adaptation via AGA. Specifically, we model the distribution of $\mathcal{A}^+$ and $\mathcal{A}^-$ by mapping them into a high-dimensional Reproducing Kernel Hilbert Space (RKHS) with Gaussian kernel $g$, and compute the regularization based on maximum mean discrepancy [42] (MMD) as follows:

$$\mathcal{R}(\mathcal{A}^+, \mathcal{A}^-) = \mathbb{E}_{m,m'\sim\mathcal{A}^+}[g(m,m')] + \mathbb{E}_{n,n'\sim\mathcal{A}^-}[g(n,n')] - 2 \cdot \mathbb{E}_{m\sim\mathcal{A}^+,n\sim\mathcal{A}^-}[g(m,n)], \quad (4)$$

where $(m, m')$ are the positive pair drawn from $\mathcal{A}^+$, and $(n, n')$ are the negative pair drawn from $\mathcal{A}^-$. The first two terms of Eq. (4) encourage similarity among positives (negatives), promoting consistency within each respective group. And the third term regularizes the model by guiding $\mathcal{A}^-$ with $\mathcal{A}^+$. Specifically, it aligns the distributions of $\mathcal{A}^-$ with those of $\mathcal{A}^+$. As a result, to minimize Eq. (4), the model has to focus on the semantically meaningful multi-modal cues from $\mathcal{A}^+$, while suppressing the knowledge from $\mathcal{A}^-$.

## 3.4 Overall Optimization

Finally, we combine Eq. (3) and Eq. (4) with a coefficient factor $\lambda$ as the overall loss function:

$$\mathcal{L}(\mathcal{X}) = \mathcal{H}(\mathcal{X}) + \lambda \cdot \mathcal{R}(\mathcal{A}^+, \mathcal{A}^-). \quad (5)$$

For the clarity of our design, we summarize PTA in Algorithm 2. When the multi-modal system receives online data $\mathcal{X}$, it encodes the data of each modality as modality-specific tokens, sends the tokens to the fusion block, and outputs the corresponding logits. We use PTA to separate $\mathcal{X}$ into $\mathcal{X}^+$ and $\mathcal{X}^-$ based on the distribution of predicted class, and then reweight their contribution using a quantile-based weighting approach. Moreover, we enforce the model to learn semantically meaningful multi-modal cues using AGA.

Table 1: Accuracy comparison with SOTAs on Kinetics50-MC for multi-modal domain shifts (severity level is 5 for all tables except Table 3 and 5). The **best** performances are highlighted.

| | Gauss. | Shot | Impul. | Defoc. | Glass | Motion | Zoom | Snow | Frost | Fog | Brit. | Contr. | Elastic | Pixel | JPEG | Avg. |
|---|---|---|---|---|---|---|---|---|---|---|---|---|---|---|---|---|
| Source | 12.92 | 13.95 | 13.05 | 37.20 | 36.94 | 45.30 | 41.79 | 30.36 | 31.88 | 20.47 | 55.29 | 18.28 | 42.30 | 38.90 | 37.77 | 31.76 |
| Tent [37] | 6.72 | 7.03 | 6.68 | 27.02 | 29.00 | 38.96 | 34.43 | 17.50 | 22.22 | 8.30 | 53.42 | 9.95 | 36.02 | 28.71 | 29.79 | 23.72 |
| ETA [25] | 12.90 | 13.80 | 13.00 | 38.83 | 39.36 | 47.42 | 43.71 | 32.50 | 33.13 | 19.90 | 57.13 | 18.07 | 44.39 | 41.48 | 39.73 | 33.02 |
| MMTTA [29] | 8.56 | 9.23 | 8.45 | 32.17 | 34.06 | 42.57 | 40.35 | 24.06 | 28.02 | 11.62 | 55.53 | 12.88 | 40.93 | 35.96 | 35.17 | 27.97 |
| ABPEM [50] | 12.27 | 13.16 | 12.24 | 40.59 | 41.08 | 50.05 | 45.92 | 33.02 | 37.21 | 19.19 | 58.41 | 20.02 | 46.25 | 40.72 | 38.55 | 33.91 |
| SuMi [9] | 12.38 | 13.51 | 12.75 | 37.21 | 36.99 | 46.11 | 42.23 | 29.80 | 31.56 | 19.10 | 55.76 | 17.92 | 41.90 | 37.78 | 36.55 | 31.44 |
| READ [43] | 14.14 | 14.96 | 14.78 | 43.12 | 41.23 | 50.12 | 45.92 | 35.06 | 37.20 | 26.28 | 58.58 | 22.09 | 46.39 | 42.97 | 38.20 | 35.40 |
| PTA | **21.93** | **22.98** | **22.11** | **47.72** | **45.92** | **52.55** | **49.31** | **40.25** | **43.57** | **39.66** | **59.99** | **27.32** | **50.35** | **50.86** | **47.59** | **41.47** |

Table 2: Accuracy comparison with SOTAs on VGGSound-MC for multi-modal domain shifts.

| | Gauss. | Shot | Impul. | Defoc. | Glass | Motion | Zoom | Snow | Frost | Fog | Brit. | Contr. | Elastic | Pixel | JPEG | Avg. |
|---|---|---|---|---|---|---|---|---|---|---|---|---|---|---|---|---|
| Source | 4.67 | 4.77 | 4.65 | 9.27 | 9.04 | 11.75 | 12.83 | 9.03 | 11.78 | 9.10 | 15.82 | 5.97 | 12.82 | 10.03 | 11.87 | 9.56 |
| Tent [37] | 0.74 | 0.74 | 0.75 | 0.96 | 0.88 | 1.16 | 1.35 | 0.90 | 1.03 | 0.79 | 1.66 | 0.74 | 1.09 | 1.11 | 1.30 | 1.01 |
| ETA [25] | 6.72 | 6.86 | 6.69 | 13.34 | 14.49 | 17.15 | 18.22 | 13.30 | 16.48 | 13.64 | 23.30 | 7.77 | 19.92 | 15.45 | 18.18 | 14.10 |
| MMTTA [29] | 2.41 | 2.79 | 2.44 | 4.16 | 3.63 | 5.24 | 5.80 | 3.63 | 3.75 | 2.14 | 8.26 | 2.14 | 4.72 | 4.76 | 6.46 | 4.14 |
| ABPEM [50] | 4.59 | 5.75 | 4.47 | 12.62 | 13.70 | 16.49 | 15.62 | 12.72 | 14.62 | 14.25 | 20.05 | 7.62 | 16.51 | 12.12 | 15.38 | 12.43 |
| SuMi [9] | 4.68 | 4.76 | 4.60 | 9.24 | 9.22 | 11.47 | 12.35 | 9.10 | 11.40 | 8.69 | 15.16 | 5.97 | 12.55 | 9.88 | 11.12 | 9.34 |
| READ [43] | 7.43 | 7.71 | 7.65 | 13.01 | 13.71 | 15.28 | 14.58 | 12.65 | 13.79 | 12.88 | 16.57 | 9.69 | 15.75 | 13.16 | 13.71 | 12.50 |
| PTA | **10.89** | **10.96** | **10.73** | **18.08** | **18.91** | **21.86** | **21.22** | **19.06** | **21.08** | **20.61** | **25.32** | **13.78** | **22.51** | **18.42** | **20.97** | **18.29** |

# 4 Experiment

## 4.1 Experimental Setting

**Datasets.** To comprehensively validate MM-TTA, we conduct experiments on benchmarks featuring both synthetic and real-world domain shifts. For synthetic shifts, we apply 15 types of corruptions [11] to the video modality and 6 types of corruptions [43] to the audio modality of Kinetics50 [14] and VGGSound [4], following the protocol in [43]. This results in a total of 90 combinations for each benchmark, *e.g.*, Kinetics50-MC and VGGSound-MC. Each corruption type is applied at 5 severity levels. For real-world domain shifts, we choose CMU-MOSI [46], CMU-MOSEI [47], and CH-SIMS [44] for evaluation, each comprising three modalities (*e.g.*, text, audio, and video). See Appendix A for more details.

**Baselines.** We compare our method with state-of-the-art (SOTA) methods, including TENT [37], ETA [25], MMTTA [29], READ [43], SuMi [9], and ABPEM [50]. Details are in Appendix C.

**Implementation details.** For experiments on synthetic multi-modal domain shifts, we adopt the pre-trained model from [43] based on the CAV-MAE architecture [8], following [43, 9, 50]. The learning rate and batch size are set to 2e-4 and 32 for Kinetics-C, and 1e-4 and 64 for VGGSound-C, respectively. For the experiments on real-world domain shifts, we provide pre-trained models for CMU-MOSI [46], CMU-MOSEI [47], and CH-SIMS [44] following the training protocol [10]. The learning rate and batch size are set to 1e-3 and 24. Details are in Appendix B. By default, hyper-parameters are set as $s = 0.5$, $\lambda = 1$. Following [43, 50], we update query/key/value transformation matrices of the attention layer in the fusion block. We run all experiments with 5 random seeds on one NVIDIA 4090 GPU, and report the average accuracy.

## 4.2 Comparison with SOTAs

**Robustness to synthetic multi-modal domain shifts.** We present the results of the most challenging task, where online data from both audio and video modalities is corrupted with the highest noise severity (level 5), in Tables 1 and 2. Each number denotes the average performance over six audio corruptions under a specific video corruption. As the rate of true positives experiences a significant drop, existing methods that employ confidence [43, 50] or entropy [25, 9] as the criterion for sample reweighting demonstrate limited improvement under multi-modal domain shifts. In comparison, PTA outperforms all existing SOTA methods across all corruption combinations. Specifically, it achieves a 9.71% and 8.73% performance gains compared to the pre-trained model on Kinetics50-MC and VGGSound-MC, respectively. This demonstrates the effectiveness of PTA in jointly assesses the contribution of each data based on their prediction confidence and bias levels, and the importance of leveraging the semantically meaningful multi-modal cues to guide model adaptation.

**Robustness to real-world multi-modal domain shifts.** We summarize the empirical results on real-world multi-modal domain shifts in Table 3, where "A→B" denotes the source domain to the target domain. We observe that existing methods [43, 25, 50] that depend on confidence or entropy as the criterion to select and reweight samples bring marginal improvements compared to the pre-trained model.

Table 3: Accuracy comparison with SOTA methods on CMU-MOSI, CMU-MOSEI, and CH-SIMS. We denote CMU-MOSI, CMU-MOSEI, and CH-SIMS as MOSI, MOSEI, and SIMS for clarity.

| | MOSI→MOSEI | MOSI→SIMS | Avg. | MOSEI →MOSI | MOSEI→SIMS | Avg. | SIMS→MOSI | SIMS→MOSEI | Avg. |
|---|---|---|---|---|---|---|---|---|---|
| Source | 54.42 | 24.80 | 39.61 | 77.00 | 30.05 | 53.53 | 51.86 | 31.42 | 41.64 |
| Tent [37] | 51.15 | 23.50 | 37.33 | 76.72 | 30.15 | 53.44 | 52.04 | 29.11 | 40.58 |
| ETA [25] | 55.04 | 25.31 | 37.33 | 77.66 | 31.05 | 54.34 | 52.32 | 32.67 | 42.45 |
| MMTTA [29] | 54.15 | 25.80 | 39.98 | 76.41 | 30.16 | 53.29 | 52.24 | 29.28 | 40.76 |
| ABPEM [50] | 56.72 | 23.80 | 40.30 | 76.83 | 31.25 | 54.04 | 52.10 | 35.86 | 43.98 |
| SuMi [9] | 56.37 | 24.40 | 40.39 | 78.11 | 30.62 | 54.37 | 52.53 | 36.36 | 44.45 |
| READ [43] | 55.63 | 24.00 | 39.82 | 76.97 | 29.80 | 53.39 | 52.10 | 35.32 | 43.71 |
| PTA | **57.17** | **25.96** | **41.66** | **78.83** | **31.85** | **55.34** | **52.94** | **38.01** | **45.48** |

Table 4: Comparisons on the key components of PTA. "Baseline + PTA$^-$" leverages only $\mathcal{X}^+$.

| | Gauss. | Shot | Impul. | Defoc. | Glass | Motion | Zoom | Snow | Frost | Fog | Brit. | Contr. | Elastic | Pixel | JPEG | Avg. |
|---|---|---|---|---|---|---|---|---|---|---|---|---|---|---|---|---|
| Baseline | 9.03 | 9.72 | 9.07 | 32.52 | 34.03 | 43.16 | 40.08 | 24.46 | 27.97 | 12.87 | 55.04 | 13.77 | 40.40 | 34.65 | 34.88 | 28.11 |
| +PTA$^-$ | 10.77 | 11.69 | 11.16 | 43.74 | 41.80 | 50.04 | 46.94 | 34.18 | 39.11 | 21.04 | 58.42 | 16.95 | 47.61 | 45.76 | 43.14 | 34.82 |
| +PTA | 20.65 | 21.65 | 20.66 | 45.85 | 44.63 | 51.34 | 48.92 | 40.11 | 43.43 | 39.23 | 58.71 | 27.07 | 49.77 | 49.68 | 46.92 | 40.57 |
| +PTA+AGA | **21.93** | **22.98** | **22.11** | **47.72** | **45.92** | **52.55** | **49.31** | **40.25** | **43.57** | **39.66** | **59.99** | **27.32** | **50.35** | **50.86** | **47.59** | **41.47** |

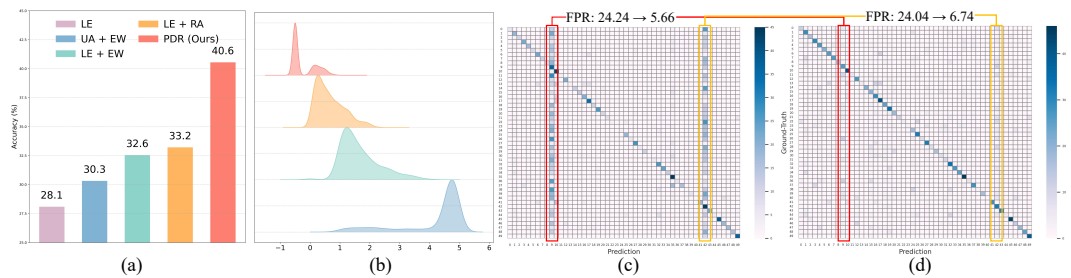

Figure 3: (a) Comparisons on existing sample reweighting methods. (b) The distribution of the weighting factors computed by comparison methods. The prediction confusion matrix of the (c) pre-trained model and (d) PDR on video corruption "Defocus" and audio corruption "Crowd".

This is because their selected "reliable" samples contain a plenty of false positives, which negates the potential improvement. In comparison, PTA outperforms all SOTA methods, and improve the pre-trained model by 2.56% on average. These improvements are mainly attributed to the effective handling of sample reliability under multi-modal domain shifts. These results demonstrate the rational design and effectiveness of PTA.

### 4.3 Ablation Study

All experiments in this subsection are performed on Kinetics50-MC (severity level 5).

**Contribution of each component.** To analyze the importance of each component, we evaluate the baseline with the proposed partition and debiased Reweighting (PDR) and multi-modal attention-guided alignment (AGA) in Table 4. The baseline is Tent [37], modified to update the same tunable parameters as ours. The results indicate that PDR is an effective sample reweighting solution under multi-modal domain shifts, indicating that the joint assessment of prediction bias and confidence levels are important. The combination of baseline and PDR results in further improvements, supporting the rationale that simultaneously promoting the knowledge in reliable samples and restraining the error propagation of unreliable ones effectively aids model adaptation. Moreover, the integration with AGA consistently improves the performance, demonstrating that guiding the optimization of $\mathcal{A}^-$ with the semantically meaningful multi-modal cues from $\mathcal{A}^+$ is effective.

**Comparison with sample reweighting methods.** To verify the effectiveness of PDR, we compare it with SOTA sample reweighting methods in MM-TTA, including low entropy (LE) [43] and single-modal assistance (UA) [9] with entropy-based reweighing (EW) [9, 25] and robust adaptation (RA) [43]. The results are summarized in Fig. 3 (a). We observe that sample reweighting-based methods outperform those relying solely on sample selection (*i.e.*, "LE"), indicating that they partially mitigate the prediction bias issue. Unlike previous methods that follow single-modal TTA methods [9] and rely solely on confidence or entropy as smaple selection and reweighting signal, PDR introduces a comprehensive measurement. Empirical evidence supports that PDR is effective in combats prediction bias under multi-modal domain shifts, as it simultaneously leverages the respective strengths of both reliable and unreliable samples.

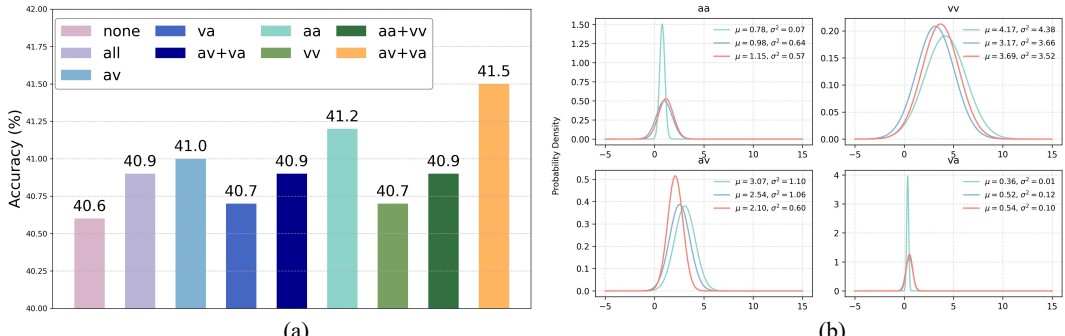

(a)                                                              (b)

Figure 4: (a) Comparisons on variants of AGA. "none" denotes the performance with the absence of AGA. (b) The distribution of attention maps for four blocks on clean, $\mathcal{A}^+$, and $\mathcal{A}^-$, respectively.

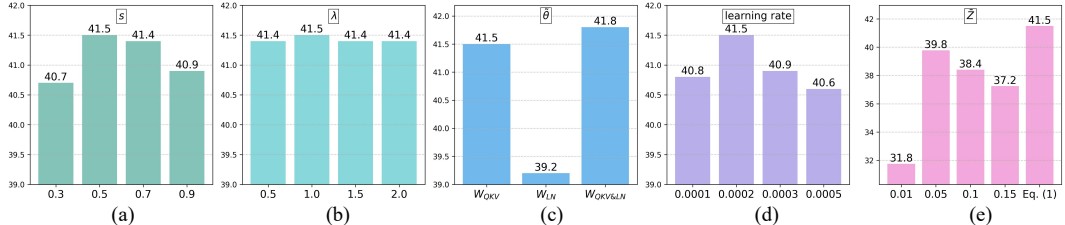

(a)                    (b)                    (c)                    (d)                    (e)

Figure 5: Comparisons on (a) $s$, (b) $\lambda$, (c) $\tilde{\Theta}$, and (d) learning rate. (e) Analysis on $\bar{Z}$.

**Analysis of PDR.** We further conduct a deeper analysis of PDR by visualizing the distribution of weighting factors for three SOTA methods in Fig. 3 (b), and providing the prediction confusion matrix for both the pre-trained model and PDR in Fig. 3 (c). As shown in Fig. 3 (b), "UA + EW" assigns high weights to the selected candidates, regardless of whether they are false positives or not. This increases the risk of error accumulation. As we demonstrate in Fig. 1 (c), the rates of true positives in predictions run low under multi-modal domain shifts, and the increasing false positives may counteract the positive contribution. "LE + EW" and "LE + RA" mitigate this issue by conservatively assigning lower weighting factors for each sample. However, this brings limited performance gains as shown in Fig. 3 (a). In contrast to these methods, PDR explicitly assigns negative weights to potential false positives, while also promoting the importance of potential true positives based on a joint assessment of prediction bias and confidence levels. This guarantees a significant improvement compared to previous methods.

As illustrated in Fig. 3 (c), the pre-trained model experience severe prediction bias dealing multi-modal domain shits. This bias can be substantially alleviated by equipping the pre-trained model with PDR, which is proven by Fig. 3 (d). Specifically, the false positive rates for the top two biased classes are 24.24 and 24.04, respectively, whereas PDR reduces them to 5.66 and 6.74. This confirms that the design of PDR mitigating false positives is effective.

We also compare our choice of $\bar{Z}$ with static values in Fig. 5 (e). We note that static values are not ideal, as overly large or small values can introduce noise. The results demonstrate $\bar{Z}$ is appropriate for representing the average bias level within the batch.

**Analysis of AGA.** To investigate the mechanism of AGA, we compare the performance for different attention blocks and their combinations in Fig. 4 (a), and provide the distribution of attention maps for each block computed by clean, $\mathcal{A}^+$, and $\mathcal{A}^-$. The attention map (denoted as "all") can be divided into self-attention blocks (denoted as "aa" and "av") and cross-attention blocks (denoted as "av" and "vv"). From Fig. 4 (a), we observe that the performance gain achieved by applying AGA on different attention blocks or their combination varies. In specific, "aa" and "va" achieve the best results for single attention block, while "vv" and "av" achieve modest improvement. As evidenced in Fig. 4 (b), the attention maps "aa" and "va" in $\mathcal{A}^+$ have distributions closer to the clean versions, whereas "vv" and "av" show distributions farther from them. This explains the necessity of guiding the optimization of $\mathcal{A}^-$ with the semantically meaningful multi-modal cues in $\mathcal{A}^+$. In practice, one can effectively choose the attention blocks with semantically meaningful multi-modal cues by analyzing the variance of each block since the optimal ones are less sensitive to multi-modal domain shifts.

**Analysis of hyper-parameters.** We summarize the ablation for hyper-parameters, including $s$ in Eq. (2), $\lambda$ in Eq. (5), the learning rate, and the tunable parameters $\tilde{\Theta}$, in Fig. 5 (a)-(d), respectively. We

Table 5: Accuracy comparison with SOTA methods on Kinetics-MC for multi-modal domain shifts. "A(X)-V(Y)" (X, Y $\in \{1, 2, 3, 4, 5\}$) denotes the noise severity level of audio and video modalities.

| | A(1)-V(1) | A(2)-V(2) | A(3)-V(3) | A(4)-V(4) | A(3)-V(5) | A(5)-V(3) | A(4)-V(5) | A(5)-V(4) | ③ | ④ |
|---|---|---|---|---|---|---|---|---|---|---|
| Source | 61.97 | 54.84 | 48.76 | 40.32 | 33.16 | 47.98 | 32.56 | 39.60 | 47.59 | 48.12 |
| Tent [37] | 59.77 | 49.00 | 41.12 | 32.51 | 23.66 | 41.84 | 23.53 | 32.44 | 41.05 | 40.51 |
| ETA [25] | 64.37 | 57.32 | 51.29 | 42.13 | 34.85 | 49.87 | 34.09 | 41.12 | 49.69 | 50.02 |
| MMTTA [29] | 61.75 | 53.28 | 46.07 | 36.92 | 28.77 | 46.26 | 28.40 | 36.49 | 45.14 | 45.49 |
| ABPEM [50] | 66.35 | 60.21 | 54.85 | 46.14 | 38.86 | 53.20 | 38.34 | 45.13 | 53.32 | 53.46 |
| SuMi [9] | 62.84 | 55.16 | 48.94 | 40.69 | 32.04 | 47.64 | 31.57 | 38.88 | 47.88 | 48.12 |
| READ [43] | 66.57 | 61.17 | 55.51 | 45.12 | 40.80 | 53.87 | 39.82 | 46.34 | 53.64 | 53.88 |
| PTA | **66.88** | **61.72** | **57.07** | **49.83** | **43.75** | **55.39** | **42.71** | **48.84** | **55.30** | **55.56** |

Table 6: Accuracy comparison with SOTA methods on Kinetics50-C with corrupted video.

| | Gauss. | Shot | Impul. | Defoc. | Glass | Motion | Zoom | Snow | Frost | Fog | Brit. | Contr. | Elastic | Pixel | JPEG | Avg. |
|---|---|---|---|---|---|---|---|---|---|---|---|---|---|---|---|---|
| Source | 48.84 | 49.36 | 48.92 | 67.47 | 61.38 | 70.83 | 66.19 | 61.18 | 61.26 | 45.35 | 76.04 | 51.64 | 65.99 | 68.55 | 65.71 | 60.58 |
| Tent [37] | 48.57 | 49.08 | 48.85 | 67.60 | 62.18 | 71.96 | 67.62 | 63.23 | 61.46 | 23.31 | 75.88 | 50.15 | 68.83 | 69.83 | 66.59 | 59.68 |
| ETA [25] | 49.43 | 50.16 | 49.73 | 67.71 | 64.18 | 71.60 | 67.96 | 63.46 | 63.14 | 49.72 | 76.13 | 52.05 | 68.48 | 69.62 | 67.22 | 62.04 |
| MMTTA [29] | 48.60 | 49.38 | 48.86 | 67.86 | 61.70 | 71.41 | 66.39 | 62.32 | 36.58 | 75.95 | 51.43 | 68.08 | 70.05 | 66.86 | 60.56 |
| ABPEM [50] | 50.25 | 51.32 | 50.59 | 67.64 | 65.37 | 71.96 | 68.10 | 63.93 | 65.52 | 60.99 | 76.08 | 52.33 | 68.98 | 69.30 | 67.75 | 63.34 |
| SuMi [9] | 49.23 | 49.59 | 49.52 | 67.50 | 62.15 | 71.08 | 66.76 | 61.65 | 61.58 | 46.39 | 76.07 | 51.64 | 66.96 | 68.36 | 66.64 | 61.01 |
| READ [43] | 51.91 | 52.28 | 51.77 | 67.95 | 65.82 | 71.63 | 69.08 | 64.81 | 65.89 | 61.67 | **76.37** | 54.21 | 69.36 | 70.03 | **68.46** | 64.08 |
| PTA | **52.93** | **53.00** | **52.48** | **68.51** | **66.87** | **72.36** | **69.11** | **64.86** | **67.11** | **64.14** | 75.92 | **55.05** | **69.15** | **70.23** | 68.41 | **64.68** |

Table 7: Comparison on Kinetics50-C with corrupted audio.

| | Gauss. | Traff. | Crowd. | Rain | Thund. | Wind | Avg. |
|---|---|---|---|---|---|---|---|
| Source | 73.88 | 65.38 | 67.87 | 69.99 | 68.39 | 70.39 | 69.32 |
| Tent [37] | 74.08 | 68.60 | 70.08 | 70.67 | 67.00 | 71.31 | 70.29 |
| ETA [25] | 74.12 | 67.95 | 69.79 | 70.90 | 70.38 | 70.91 | 70.68 |
| MMTTA [29] | **74.36** | 67.31 | 69.53 | **70.91** | 68.77 | **70.95** | 70.31 |
| ABPEM [50] | 74.12 | 69.20 | 69.84 | 70.68 | 72.32 | 70.40 | 71.09 |
| SuMi [9] | 73.87 | 66.19 | 68.28 | 70.38 | 70.58 | 70.00 | 69.88 |
| READ [43] | 74.12 | 69.30 | 70.02 | 70.57 | 72.64 | 70.85 | 71.25 |
| PTA | 73.52 | **70.00** | **70.40** | 70.31 | **73.36** | 70.79 | **71.40** |

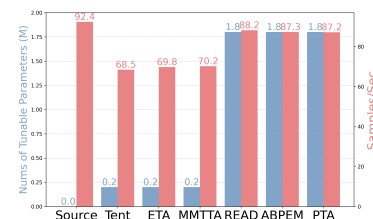

Figure 6: Efficiency comparisons.

denote the query/key/value transformation matrices of the attention layer and the Layer Normalization (LayerNorm) module in the pre-trained model as $W_{QKV}$ and $W_{LN}$, respectively. Fig. 5 (a)-(d) demonstrate the robustness of our method to hyper-parameters.

## 4.4 Further Analysis

**Exploration on changing environment.** Considering that the noise severity levels of each modality may fluctuate in dynamic environments, we further categorize multi-modal domain shifts into four distinct cases (See Appendix G): ① video and audio noise levels change synchronously; ② video noise level is constant, audio noise varies independently; ③ video noise level varies randomly, audio noise level matches it synchronously; ④ both video and audio noise levels vary randomly and independently. The results are summarized in Table 5. We observe that the performance of the pre-trained model ("Source") varies substantially across ①-④, highlighting the importance of assessing the impact of varying noise severity levels in each modality. Specifically, for ①-④, PTA achieves the highest performance, with average improvements of 7.4%, 9.3%, 7.7%, and 7.4% over the pre-trained model, respectively. These empirical results demonstrate the effectiveness and robustness of our method in dynamic environments, delivering the best performance under changing environment.

**Comparison on single-modal domain shifts.** We evaluate the performance of comparison methods under single-modal domain shifts, with the results for corrupted video and audio presented in Tables 6 and 7, respectively. Our method improves performance by 4.1% and 1.1% on corrupted video and audio modalities, respectively, over the pre-trained model. This demonstrate PTA's robustness to both single- and multi-modal domain shifts, whereas existing methods often fail on the latter case.

**Efficiency comparisons.** Following [43, 50], we report the number of samples processed per second (samples/sec) and the total count of trainable parameters, *i.e.*, in millions (M), to enable efficiency comparisons in Fig. 6. In summary, our method exhibits similar efficiency compared to SOTA methods.

# 5    Conclusions

In this paper, we investigate a practical challenge: multi-modal domain shifts for TTA. We reveal that existing single-modal methods struggle to identify and utilize reliable samples within a batch amid severe prediction bias. To address this challenge, we propose Partition-Then-Adapt (PTA), which comprises Partition and Debiased Reweighting (PDR) and multi-modal Attention-Guided Alignment (AGA). PDR partitions online data into potential reliable and unreliable subsets based on prediction bias, assessed via the uniformity of their predicted label distribution. It then employs a quantile-based reweighting strategy that adjusts sample contributions in entropy optimization by jointly considering prediction bias and confidence levels. AGA further regularizes PDR to focus on semantically meaningful multi-modal cues by aligning the attention map distributions of unreliable subsets with those of reliable ones through maximum mean discrepancy regularization. Extensive experiments demonstrate the effectiveness of PTA, addressing both single- and challenging multi-modal domain shifts for TTA in multi-modal tasks.

# 6    Acknowledgement

This work was supported by the National Natural Science Foundation of China under Grant 62476099, 62076101 and 62406323, the Postdoctoral Fellowship Program of CPSF (No. GZC20232993), the China Postdoctoral Science Foundation (No. 2024M753496), and Guangdong Basic and Applied Basic Research Foundation under Grant 2024B1515020082 and 2023A1515010007, the Guangdong Provincial Key Laboratory of Human Digital Twin under Grant 2022B1212010004, the TCL Young Scholars Program.

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

# Appendix

This appendix provides additional results, figures, and graphs to better illustrate our method. Specifically, we provide detailed information on the benchmark, the pre-trained model, and the implementation in Sections A, B, and C, respectively. Additionally, Section D presents further experimental results, including detailed main experiment and comparisons under continual settings. Section E explores different design variants of our method. Section F provides the theoretical analysis of PDR. We also include illustrations of changing environments with varying severity levels, attention maps, and confusion matrices to facilitate intuitive understanding in G. Finally, we discuss the limitations and broader impacts of our work in Sections H and I, respectively.

## A   Benchmark Details

We construct two benchmarks based on Kinetics50 [14] and VGGSound [4], to evaluate the performance of SOTA methods under synthetic multi-modal domain shifts during test-time adaptation. The examples are shown in Figure 7. Moreover, we test the comparison methods for real-world multi-modal domain shifts on CMU-MOSI [46], CMU-MOSEI [47], and CH-SIMS [44].

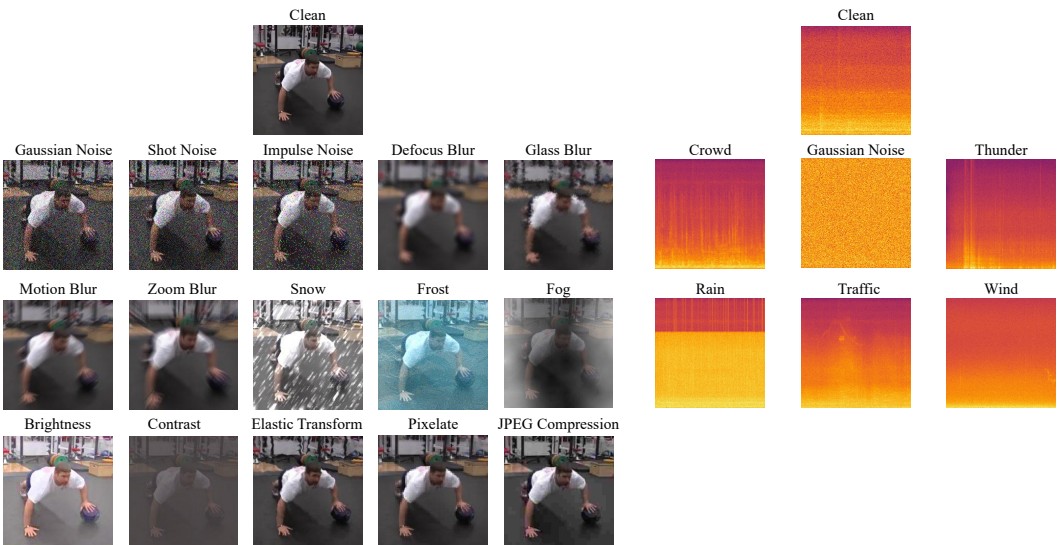

Figure 7: Examples for corrupted video and audio data.

**Kinetics.** The Kinetics dataset is a comprehensive and high-quality benchmark designed for recognizing human actions in videos. It contains approximately 500,000 video clips that span 600 distinct action classes, with each class having at least 600 clips. Each video clip is around 10 seconds long and is associated with a single action label. The videos were sourced from YouTube. In line with the previous works [43, 9, 50], we focus on a subset of the Kinetics dataset, which includes 50 action classes, comprising 2,466 test pairs.

**VGGSound.** The VGGSound dataset is a large-scale audio-visual correspondence benchmark consisting of short audio clips extracted from videos uploaded to YouTube. All videos are captured "in the wild", ensuring that there is a clear correspondence between the audio and visual content, meaning that the sound source is visually identifiable. Each video in this benchmark has a fixed duration of 10 seconds. Following privious work [43, 9, 50], we obtain 14,046 testing visual-audio pairs.

**Kinetics50-MC and VGGSound-MC.** Following [43, 9, 50], we introduce 15 types of synthetic noise [11], including "Gaussian Noise", "Shot Noise", "Impulse Noise", "Defocus Blur", "Glass Blur", "Motion Blur", "Zoom Blur", "Snow", "Frost", "Fog", "Brightness", "Contrast", "Elastic Transform", "Pixelate", and "JPEG Compression", to corrupt the video modality data. For the audio modality data, we introduce 1 type of synthetic noise and 5 types of real-world noise, comprising "Gaussian Noise", "Paris Traffic Noise", "Crowd Noise", "Rainy Noise", "Thunder Noise" and "Windy Noise". Each corruption type is applied at five levels of severity. In this way, a total of 90 types of corruption combinations are prepared for Kinetics50-MC and VGGSound-MC.

**CMU-MOSI.** CMU-MOSI [46] benchmark is a widely used benchmark for multimodal sentiment analysis, comprising 2,199 short video clips from YouTube. Each clip includes aligned text (transcripts), audio, and facial visual data, annotated with sentiment intensity (ranging from -3 to +3) and binary labels.

**CMU-MOSEI.** CMU-MOSEI [47] benchmark is a large-scale benchmark for multimodal sentiment and emotion analysis, containing 23,454 video clips from YouTube. Each clip includes synchronized text, audio, and visual modalities, annotated with sentiment scores (ranging from -3 to +3) and six discrete emotions (happiness, sadness, anger, fear, disgust, surprise).

**CH-SIMS.** CH-SIMS [44] is a multimodal sentiment analysis dataset for mandarin, featuring 2,281 video clips from Chinese TV shows and vlogs. It provides text, audio, and visual data, annotated with both continuous sentiment scores (ranging from -1 to +1) and binary labels.

For sentiment recognition, as three datasets has different emotion labels, we categorize continuous sentiment scores into three classes: scores less than 0 as "Negative" scores equal to 0 as "Neutral", and scores greater than 0 as "Positive". In this way, we are able to test the model, which is pre-trained on the source domain, on the target domain. We adopt the pre-processed data of CMU-MOSI, CMU-MOSEI, and CH-SIMS provided by [10].

# B Pre-trained Model Details

For the experiments on synthetic multi-modal domain shifts, we use the pre-trained CAV-MAE model provided by [43], consistent with previous works [9, 50]. Specifically, the model is pre-trained on the respective training sets of Kinetics and VGGSound, respectively. In other words, Kinetics50 and VGGSound serve as the source domains, while Kinetics50-MC and VGGSound-MC represent the target domains.

For the experiments on real-world multi-modal domain shifts, we train three models separately for the CMU-MOSI [46], CMU-MOSEI [47], and CH-SIMS [44] benchmarks. An Adam optimizer with a learning rate of 1e-4 is adopted. The batch size is set to 24, and each model is trained for 30 epochs. We use the validation set of each dataset to select the best pre-trained model.

# C Implementation Details

**TENT**. For TENT [37], we use the Adam optimizer with a learning rate of 1e-4 on Kinetics-MC and VGGSound-MC. We set the tunable parameters as those in LayerNorm Module. The implementation follows the official code[2].

**ETA**. For ETA [25], we use the Adam optimizer with a learning rate of 1e-4 on Kinetics-MC and VGGSound-MC. We set the tunable parameters as those in LayerNorm module. Moreover, we set the exponential moving average factor, the cosine similarity threshold, and the entropy threshold to 0.9, 0.05, and $0.4 \times \ln(\mathcal{C})$, respectively. Here, $\mathcal{C}$ is the number of classes. The implementation follows the official code[3].

**MMTTA**. For MMTTA [29], We use the Adam optimizer with a learning rate of 1e-4 on Kinetics-MC and VGGSound-MC. We set the tunable parameters as those in LayerNorm module. Moreover, we set the exponential moving average factor for the teacher model as 0.995. The implementation follows the official code[4].

**READ**. For READ [43], we use the Adam optimizer with a learning rate of 1e-4 on Kinetics-MC and VGGSound-MC. The tunable parameters are set to the query/key/value transformation matrices of the attention layer in the fusion block. The implementation follows the official code[5].

**SuMi**. For SuMi [9], we use the Adam optimizer with a learning rate of 1e-4 and 1e-5 on Kinetics-MC and VGGSound-MC, respectively. The tunable parameters are set to the query/key/value transformation matrices of the attention layer in the fusion block. The implementation follows the official code[6].

**ABPEM**. For ABPEM [9], we use the Adam optimizer with a learning rate of 1e-4 on Kinetics-MC and VGGSound-MC. The tunable parameters are set to the query/key/value transformation matrices

---

[2]https://github.com/DequanWang/tent

[3]https://github.com/mr-eggplant/EATA

[4]https://www.nec labs.com/~mas/MM-TTA

[5]https://github.com/XLearning-SCU/2024-ICLR-READ

[6]https://github.com/zrguo/SuMi

of the attention layer in the fusion block. We re-implement ABPEM based on their original paper since the code is not public available.

# D  More Experimental Results

**Main experiment details.** We report the detailed results of comparison methods on total 90 corruption combinations, and summarize the results in Table 10. In summary, our method achieves the best results across all corruption combinations, significantly outperforms the SOTA methods.

**Comparison on continual settings.** Continual test-time adaptation [40] (CTTA) introduce a challenging setting, where a pre-trained model operates in non-stationary and continuously changing environments, with the target domain distribution changing over time. All previous methods do not consider this setting; instead, they focus on a fully test-time adaptation (FTTA) [37] scenario, where the pre-trained model can restore its parameters when distribution shift changes. We test our method and SOTAs on the CTTA setting with multi-modal domain shifts, and report their performance in Table 8. We observe that most methods fail to handle CTTA, mostly because the accumulated errors render catastrophic forgetting [40]. In comparison, ETA [25] and PTA still outperform the pre-trained model with different adaptation strategies. Specifically, ETA [25] limits the data used in loss computation to a very small subset, which means the pre-trained model could be unchanged when the online data contains extensive noise. On the other hand, PTA effectively handles sample reliability, preventing error accumulation, and eventually outperforms the pre-trained model by 11.9%.

**Comparison on different batch sizes.** In real-world scenarios, online data is often insufficient, and at times, only a limited amount is available, posing significant challenges for effective model adaptation. Previous research also indicates that batch size is a key factor influencing overall performance during TTA [26]. Therefore, we validate our method with different batch sizes, and summarize the performance in Fig. 8. To adapt to extremely small batch sizes, we set a memory queue to store history predictions to compute Eq. (1) and Eq. (2). In other words, $\mathcal{Z}$ and $\mathcal{K}$ are transformed from an online manner to an offline manner in this case. From Fig. 8, we observe that most methods fail to handle small batch sizes, because their weighting factors become severely unreliable. On the contrary, our method shows robustness to small batch sizes, consistently outperforms the pre-trained model ("Source") by 7.2% on average.

**Comparison on CIFAR-10/100-C and ImageNet-C.** We run experiments on single-modal datasets, such as CIFAR-10/100-C [12] and ImageNet-C [12] using CNN-based backbones. We report the average accuracy over 15 types of corruptions for these benchmarks in Table 13. We observe that our method outperforms READ on single-modal datasets and matches specialized single-modal methods (surpassing on ImageNet-C). Crucially, it excels over all SOTAs in challenging multi-modal domain shifts.

**Comparison on multi-modal domain shifts using CLIP.** We also run experiments on CLIP (ViT-B-16) [27]. To simulate multi-modal domain shifts, we use ImageNet-C [12] for the image branch and apply Gaussian Noise to the text branch. The results are presented in Table 14. These results indicate that our method outperforms others in the majority of cases by a substantial margin, thereby confirming its effectiveness on vision-language models.

Table 8: Accuracy comparison with SOTA methods on Kinetics50-MC with corrupted video and audio modalities (severity level 5) in continual settings.

| | Gauss. | Shot | Impul. | Defoc. | Glass | Motion | Zoom | Snow | Frost | Fog | Brit. | Contr. | Elastic | Pixel | JPEG | Avg. |
|---|---|---|---|---|---|---|---|---|---|---|---|---|---|---|---|---|
| Tent [37] | 6.72 | 2.21 | 1.97 | 1.99 | 1.97 | 1.97 | 1.97 | 1.97 | 2.07 | 1.97 | 1.97 | 1.97 | 1.97 | 1.97 | 2.07 | 2.32 |
| ETA [25] | 12.90 | 13.29 | 12.43 | 36.39 | 42.10 | 51.07 | 49.60 | 36.51 | 40.57 | 36.74 | 57.25 | 26.86 | 54.89 | 47.88 | 44.73 | 37.55 |
| SAR [26] | 12.18 | 6.34 | 5.56 | 5.34 | 6.17 | 4.05 | 3.44 | 3.16 | 3.08 | 2.76 | 2.55 | 2.23 | 1.96 | 1.96 | 1.96 | 4.18 |
| READ [43] | 14.14 | 24.33 | 23.83 | 37.86 | 35.27 | 33.95 | 29.64 | 19.69 | 21.33 | 19.45 | 34.30 | 12.49 | 24.79 | 21.99 | 19.65 | 24.85 |
| SuMi [9] | 12.38 | 6.63 | 2.67 | 3.62 | 3.37 | 2.93 | 2.83 | 2.51 | 2.48 | 2.15 | 2.67 | 2.08 | 2.77 | 2.20 | 2.47 | 3.58 |
| ABPEM [50] | 12.27 | 12.65 | 12.71 | 29.51 | 29.00 | 33.82 | 32.47 | 25.57 | 28.82 | 26.04 | 36.70 | 17.78 | 33.72 | 32.07 | 30.16 | 26.22 |
| PTA | **20.55** | **26.38** | **27.72** | **47.78** | **45.20** | **52.63** | 48.22 | **38.16** | **42.38** | 32.12 | 53.38 | 25.41 | 44.70 | 44.48 | 39.48 | **39.24** |

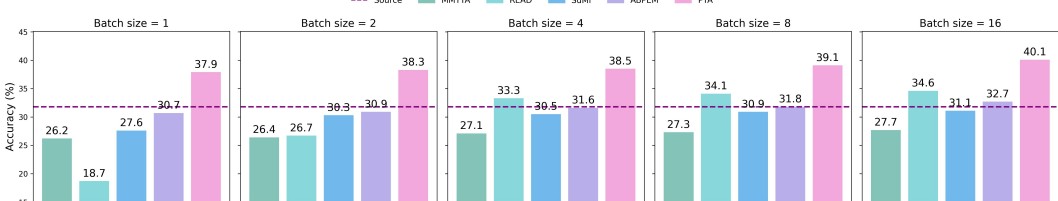

Figure 8: Performance comparisons on different batch sizes.

Table 9: Accuracy comparison with the original and variants (denoted as *) of PDR on Kinetics50-MC with corrupted video and audio modalities (severity level 5).

| | Gauss. | Shot | Impul. | Defoc. | Glass | Motion | Zoom | Snow | Frost | Fog | Brit. | Contr. | Elastic | Pixel | JPEG | Avg. |
|---|---|---|---|---|---|---|---|---|---|---|---|---|---|---|---|---|
| Source | 12.92 | 13.95 | 13.05 | 37.20 | 36.94 | 45.30 | 41.79 | 30.36 | 31.88 | 20.47 | 55.29 | 18.28 | 42.30 | 38.90 | 37.77 | 31.76 |
| PDR*+AGA | 21.78 | 23.18 | **22.35** | 46.35 | 45.37 | 51.07 | 48.57 | 39.08 | 42.93 | 38.90 | 59.02 | **27.36** | 49.17 | 49.63 | 46.31 | 40.74 |
| PDR+AGA | **21.93** | **22.98** | 22.11 | **47.72** | **45.92** | **52.55** | **49.31** | **40.25** | **43.57** | **39.66** | **59.99** | 27.32 | **50.35** | **50.86** | **47.59** | **41.47** |

Table 10: Accuracy comparison with SOTA methods on Kinetics50-MC with corrupted video and audio modalities (severity level 5). "Audio-c" denotes the corruption type of audio modality. The **best** performances are highlighted.

| Audio-c:Crowd | Gauss. | Shot | Impul. | Defoc. | Glass | Motion | Zoom | Snow | Frost | Fog | Brit. | Contr. | Elastic | Pixel | JPEG | Avg. |
|---|---|---|---|---|---|---|---|---|---|---|---|---|---|---|---|---|
| Source | 9.34 | 9.54 | 9.35 | 31.97 | 35.45 | 41.31 | 38.78 | 28.37 | 29.90 | 19.99 | 54.03 | 16.09 | 41.16 | 36.97 | 38.47 | 29.38 |
| Tent [37] | 4.83 | 5.07 | 5.15 | 13.60 | 35.53 | 25.86 | 28.38 | 16.74 | 22.32 | 8.06 | 55.45 | 9.05 | 41.32 | 22.48 | 26.88 | 21.38 |
| ETA [25] | 8.86 | 9.13 | 9.18 | 34.53 | 38.02 | 45.25 | 41.52 | 30.75 | 31.63 | 19.49 | 56.50 | 15.65 | 43.53 | 40.01 | 41.16 | 31.01 |
| MMTTA [29] | 6.44 | 6.81 | 6.27 | 22.86 | 36.47 | 37.32 | 39.36 | 23.92 | 28.04 | 11.44 | 55.51 | 10.81 | 42.77 | 35.93 | 37.71 | 26.78 |
| READ [43] | 8.61 | 8.43 | 8.87 | 40.36 | 38.88 | 47.88 | 42.91 | 32.60 | 35.74 | 26.03 | 57.91 | 19.09 | 44.64 | 42.73 | 42.15 | 33.12 |
| SuMi [9] | 8.66 | 9.21 | 9.03 | 32.66 | 35.97 | 43.34 | 39.39 | 28.65 | 29.88 | 19.33 | 54.38 | 16.16 | 40.69 | 37.03 | 38.81 | 29.55 |
| ABPEM [50] | 7.20 | 7.10 | 6.68 | 39.98 | 38.95 | 47.70 | 42.93 | 32.60 | 35.74 | 21.58 | 57.65 | 17.87 | 44.57 | 42.72 | 42.08 | 32.36 |
| PTA | **20.02** | **20.74** | **19.75** | **45.90** | **43.50** | **51.16** | **47.13** | **38.98** | **42.03** | **38.59** | **59.32** | **26.29** | **49.13** | **49.57** | **46.39** | **39.90** |

| Audio-c:Gaussian | Gauss. | Shot | Impul. | Defoc. | Glass | Motion | Zoom | Snow | Frost | Fog | Brit. | Contr. | Elastic | Pixel | JPEG | Avg. |
|---|---|---|---|---|---|---|---|---|---|---|---|---|---|---|---|---|
| Source | 17.96 | 19.33 | 18.36 | 47.56 | 40.97 | 54.29 | 48.48 | 32.71 | 35.67 | 21.94 | 60.17 | 24.62 | 43.98 | 48.26 | 44.26 | 37.24 |
| Tent [37] | 11.70 | 11.96 | 11.60 | 45.76 | 24.34 | 54.14 | 50.29 | 14.04 | 20.81 | 8.74 | 60.95 | 13.45 | 25.03 | 48.13 | 37.96 | 29.26 |
| ETA [25] | 18.17 | 19.61 | 18.74 | 48.34 | 43.45 | 54.70 | 50.64 | 35.12 | 36.42 | 21.90 | 61.29 | 24.74 | 46.12 | 50.93 | 46.26 | 38.43 |
| MMTTA [29] | 15.07 | 16.92 | 15.58 | 47.11 | 36.96 | 54.35 | 49.39 | 20.41 | 28.47 | 12.73 | 60.92 | 18.41 | 35.65 | 48.86 | 42.43 | 33.55 |
| READ [43] | 21.02 | 23.01 | 22.03 | 50.51 | 45.35 | 56.42 | 51.94 | 39.02 | 40.91 | 25.51 | 61.65 | 29.70 | 50.31 | 52.87 | 47.60 | 41.19 |
| SuMi [9] | 17.85 | 19.26 | 18.33 | 47.44 | 40.21 | 54.76 | 48.73 | 30.22 | 35.57 | 20.67 | 60.24 | 24.15 | 42.55 | 48.57 | 44.26 | 36.85 |
| ABPEM [50] | 20.80 | 22.59 | 21.92 | 50.64 | 45.07 | 56.25 | 51.70 | 26.15 | 40.52 | 18.00 | 61.65 | 29.52 | 49.19 | 53.23 | 47.54 | 39.65 |
| PTA | **26.33** | **27.76** | **27.23** | **51.87** | **50.07** | **56.94** | **54.10** | **43.80** | **47.52** | **44.22** | **62.31** | **32.33** | **53.79** | **54.86** | **50.25** | **45.56** |

| Audio-c:Rain | Gauss. | Shot | Impul. | Defoc. | Glass | Motion | Zoom | Snow | Frost | Fog | Brit. | Contr. | Elastic | Pixel | JPEG | Avg. |
|---|---|---|---|---|---|---|---|---|---|---|---|---|---|---|---|---|
| Source | 12.96 | 14.06 | 13.32 | 40.49 | 38.04 | 48.31 | 40.70 | 29.68 | 32.68 | 19.45 | 54.51 | 14.49 | 43.35 | 39.34 | 38.89 | 32.02 |
| Tent [37] | 4.18 | 4.33 | 4.09 | 29.90 | 33.78 | 49.35 | 31.75 | 19.08 | 25.16 | 7.65 | 55.31 | 4.83 | 35.18 | 17.76 | 26.21 | 23.24 |
| ETA [25] | 12.81 | 13.96 | 13.40 | 42.21 | 39.80 | 49.80 | 42.50 | 32.13 | 33.57 | 18.48 | 56.54 | 13.52 | 45.31 | 42.36 | 40.75 | 33.14 |
| MMTTA [29] | 5.61 | 5.65 | 5.33 | 38.67 | 36.99 | 48.76 | 41.01 | 24.02 | 29.54 | 10.76 | 54.64 | 6.41 | 42.09 | 35.21 | 35.85 | 28.04 |
| READ [43] | 15.66 | 16.95 | 15.99 | 44.70 | 41.58 | 50.80 | 44.66 | 34.17 | 36.76 | 27.30 | 57.24 | 19.68 | 46.28 | 44.44 | 43.08 | 35.95 |
| SuMi [9] | 12.39 | 13.24 | 12.98 | 40.90 | 37.86 | 48.64 | 40.78 | 29.50 | 32.47 | 17.60 | 54.49 | 13.22 | 43.30 | 39.18 | 38.86 | 31.69 |
| ABPEM [50] | 15.25 | 16.52 | 15.80 | 44.46 | 41.39 | 50.37 | 44.49 | 34.10 | 36.62 | 21.30 | 56.95 | 10.97 | 46.16 | 44.79 | 43.06 | 34.82 |
| PTA | **20.26** | **21.23** | **20.08** | **46.56** | **45.48** | **51.60** | **47.91** | **38.33** | **42.28** | **38.26** | **58.21** | **24.87** | **50.04** | **50.11** | **47.24** | **40.16** |

| Audio-c:Thunder | Gauss. | Shot | Impul. | Defoc. | Glass | Motion | Zoom | Snow | Frost | Fog | Brit. | Contr. | Elastic | Pixel | JPEG | Avg. |
|---|---|---|---|---|---|---|---|---|---|---|---|---|---|---|---|---|
| Source | 14.84 | 15.59 | 14.83 | 33.41 | 35.91 | 43.22 | 42.29 | 32.27 | 32.92 | 23.73 | 53.35 | 20.00 | 42.26 | 31.91 | 29.36 | 31.06 |
| Tent [37] | 7.26 | 7.80 | 7.20 | 15.21 | 17.68 | 23.52 | 23.82 | 20.73 | 27.46 | 10.95 | 37.03 | 12.00 | 33.89 | 14.16 | 13.18 | 18.13 |
| ETA [25] | 14.76 | 15.52 | 14.47 | 35.76 | 38.68 | 45.60 | 43.79 | 33.55 | 33.83 | 23.64 | 56.15 | 20.09 | 44.87 | 34.11 | 29.99 | 32.32 |
| MMTTA [29] | 8.71 | 9.31 | 8.72 | 19.86 | 24.28 | 31.79 | 32.55 | 27.35 | 31.44 | 15.33 | 50.27 | 16.17 | 40.79 | 19.58 | 18.17 | 23.62 |
| READ [43] | 13.79 | 12.61 | 16.06 | 43.29 | 42.87 | 51.01 | 47.70 | 37.09 | 40.05 | 32.42 | 59.29 | 23.30 | 47.35 | 32.33 | 13.73 | 34.19 |
| SuMi [9] | 13.84 | 14.95 | 13.94 | 33.75 | 37.01 | 45.12 | 43.78 | 31.83 | 32.91 | 22.96 | 55.01 | 19.29 | 41.94 | 25.44 | 21.25 | 30.20 |
| ABPEM [50] | 8.43 | 9.03 | 8.37 | 28.89 | 43.03 | 51.08 | 47.95 | 37.67 | 40.17 | 31.71 | 59.03 | 22.97 | 47.58 | 17.97 | 16.18 | 31.34 |
| PTA | **24.40** | **26.08** | **24.87** | **50.91** | **49.25** | **55.33** | **51.03** | **42.90** | **45.96** | **42.20** | **62.35** | **30.58** | **53.29** | **53.70** | **50.01** | **44.19** |

| Audio-c:Traffic | Gauss. | Shot | Impul. | Defoc. | Glass | Motion | Zoom | Snow | Frost | Fog | Brit. | Contr. | Elastic | Pixel | JPEG | Avg. |
|---|---|---|---|---|---|---|---|---|---|---|---|---|---|---|---|---|
| Source | 9.46 | 10.60 | 9.58 | 29.08 | 32.72 | 37.35 | 37.53 | 25.40 | 27.58 | 18.53 | 52.61 | 13.35 | 40.14 | 32.78 | 33.62 | 27.36 |
| Tent [37] | 5.73 | 5.89 | 5.86 | 18.32 | 32.70 | 33.10 | 33.67 | 12.15 | 16.31 | 6.48 | 54.19 | 6.79 | 36.79 | 24.98 | 33.31 | 21.75 |
| ETA [25] | 9.73 | 10.68 | 9.75 | 30.35 | 35.59 | 40.79 | 39.52 | 28.30 | 29.33 | 17.57 | 54.53 | 13.51 | 41.70 | 35.60 | 36.57 | 28.90 |
| MMTTA [29] | 7.83 | 8.24 | 7.59 | 24.04 | 33.63 | 35.70 | 37.96 | 19.21 | 23.24 | 9.06 | 54.21 | 8.97 | 41.05 | 30.59 | 34.16 | 25.03 |
| READ [43] | 10.82 | 11.54 | 10.36 | 36.16 | 37.29 | 45.17 | 42.55 | 30.24 | 33.18 | 24.77 | 56.85 | 15.89 | 43.54 | 38.81 | 38.66 | 31.72 |
| SuMi [9] | 9.23 | 10.26 | 9.41 | 28.06 | 32.98 | 37.88 | 37.95 | 25.33 | 26.66 | 16.66 | 53.24 | 13.37 | 39.99 | 32.49 | 33.98 | 27.17 |
| ABPEM [50] | 8.82 | 9.98 | 8.74 | 35.73 | 37.18 | 45.22 | 42.62 | 30.28 | 33.43 | 11.62 | 56.69 | 14.69 | 43.79 | 38.93 | 38.62 | 30.42 |
| PTA | **18.67** | **19.27** | **18.86** | **44.22** | **42.20** | **49.20** | **46.79** | **37.87** | **41.01** | **35.53** | **58.38** | **23.85** | **46.77** | **47.20** | **44.58** | **38.29** |

| Audio-c:Wind | Gauss. | Shot | Impul. | Defoc. | Glass | Motion | Zoom | Snow | Frost | Fog | Brit. | Contr. | Elastic | Pixel | JPEG | Avg. |
|---|---|---|---|---|---|---|---|---|---|---|---|---|---|---|---|---|
| Source | 12.96 | 14.57 | 12.87 | 40.71 | 38.57 | 47.34 | 42.93 | 33.71 | 32.50 | 19.18 | 57.05 | 21.12 | 42.90 | 44.16 | 42.03 | 33.51 |
| Tent [37] | 6.63 | 7.10 | 6.19 | 39.34 | 29.99 | 47.80 | 38.64 | 22.23 | 21.25 | 7.94 | 57.60 | 13.57 | 43.92 | 44.76 | 41.19 | 28.54 |
| ETA [25] | 13.05 | 13.87 | 12.44 | 41.76 | 40.63 | 48.35 | 44.28 | 35.16 | 34.00 | 18.33 | 57.79 | 20.89 | 44.78 | 45.87 | 43.63 | 34.32 |
| MMTTA [29] | 7.69 | 8.43 | 7.18 | 40.47 | 36.02 | 47.48 | 41.80 | 29.46 | 27.40 | 10.40 | 57.65 | 16.48 | 43.22 | 45.61 | 42.69 | 30.80 |
| READ [43] | 14.94 | 17.23 | 15.36 | 43.67 | 41.38 | 49.41 | 45.77 | 37.22 | 36.58 | 21.62 | 58.56 | 24.85 | 46.22 | 46.66 | 43.95 | 36.23 |
| SuMi [9] | 12.30 | 14.11 | 12.79 | 40.47 | 37.92 | 47.03 | 42.74 | 33.28 | 31.88 | 17.35 | 57.20 | 21.30 | 42.93 | 43.98 | 42.13 | 33.16 |
| ABPEM [50] | 13.12 | 13.75 | 11.91 | 43.83 | 40.85 | 49.67 | 45.84 | 37.30 | 36.80 | 10.92 | 58.48 | 24.09 | 46.18 | 46.66 | 43.79 | 34.88 |
| PTA | **21.87** | **22.81** | **21.85** | **46.88** | **45.02** | **51.08** | **48.88** | **39.61** | 2.64 | **39.13** | **59.39** | **25.99** | **49.05** | **49.73** | **47.05** | **40.73** |

Table 11: Accuracy comparison with the original and three variants (denoted as MSE, L2, and KL) of AGA on Kinetics50-MC with corrupted video and audio modalities (severity level 5).

| | Gauss. | Shot | Impul. | Defoc. | Glass | Motion | Zoom | Snow | Frost | Fog | Brit. | Contr. | Elastic | Pixel | JPEG | Avg. |
|---|---|---|---|---|---|---|---|---|---|---|---|---|---|---|---|---|
| Source | 12.92 | 13.95 | 13.05 | 37.20 | 36.94 | 45.30 | 41.79 | 30.36 | 31.88 | 20.47 | 55.29 | 18.28 | 42.30 | 38.90 | 37.77 | 31.76 |
| MSE | 20.59 | 21.64 | 20.83 | 45.87 | 44.66 | 51.34 | 48.92 | 40.12 | 43.43 | 39.23 | 58.71 | 27.02 | 49.75 | 49.67 | 46.92 | 40.58 |
| L2 | 21.52 | 22.68 | 21.85 | 47.04 | 45.09 | 52.02 | 48.99 | 39.00 | 42.66 | 37.10 | 59.94 | 26.73 | 49.23 | 50.53 | 47.72 | 40.81 |
| KL | 20.59 | 21.60 | 20.85 | 45.59 | 44.61 | 51.33 | 48.90 | 40.07 | 43.39 | 39.23 | 58.74 | 27.18 | 49.75 | 49.68 | 46.90 | 40.58 |
| MMD | 21.93 | 22.98 | 22.11 | 47.72 | 45.92 | 52.55 | 49.31 | 40.25 | 43.57 | 39.66 | 59.99 | 27.32 | 50.35 | 50.86 | 47.59 | 41.47 |

Table 12: Accuracy comparison with the original and two variants (denoted as feature- and logits-level) of AGA on Kinetics50-MC with corrupted video and audio modalities (severity level 5).

| | Gauss. | Shot | Impul. | Defoc. | Glass | Motion | Zoom | Snow | Frost | Fog | Brit. | Contr. | Elastic | Pixel | JPEG | Avg. |
|---|---|---|---|---|---|---|---|---|---|---|---|---|---|---|---|---|
| Source | 12.92 | 13.95 | 13.05 | 37.20 | 36.94 | 45.30 | 41.79 | 30.36 | 31.88 | 20.47 | 55.29 | 18.28 | 42.30 | 38.90 | 37.77 | 31.76 |
| Feature-level | 20.57 | 21.57 | 20.94 | 45.87 | 44.64 | 51.35 | 48.95 | 40.13 | 43.40 | 39.38 | 58.74 | 27.05 | 49.77 | 49.58 | 46.96 | 40.59 |
| Feature-map-level | 20.66 | 21.71 | 20.66 | 45.88 | 44.65 | 51.34 | 48.93 | 40.10 | 43.41 | 39.22 | 58.74 | 26.92 | 49.75 | 49.68 | 46.94 | 40.57 |
| Logits-level | 20.44 | 21.61 | 20.55 | 45.58 | 44.87 | 51.22 | 48.74 | 39.55 | 43.29 | 39.34 | 56.76 | 26.88 | 49.74 | 49.09 | 46.74 | 40.29 |
| Attention-level | 21.93 | 22.98 | 22.11 | 47.72 | 45.92 | 52.55 | 49.31 | 40.25 | 43.57 | 39.66 | 59.99 | 27.32 | 50.35 | 50.86 | 47.59 | 41.47 |

Table 13: Comparison on CIFAR10-C, CIFAR100-C, and ImageNet-C.

| Method | CIFAR-10-C | CIFAR-100-C | ImageNet-C |
|---|---|---|---|
| Source | 56.5 | 53.6 | 18.0 |
| TENT [37] | 81.7 | 68.5 | 34.7 |
| ETA [25] | **81.9** | **68.9** | 39.7 |
| READ [43] | 79.0 | 61.5 | 31.3 |
| Ours | 81.3 | 68.8 | **41.3** |

Table 14: Accuracy comparison using CLIP.

| Method | Gauss. | Shot | Impul. | Defoc. | Glass | Motion | Zoom | Snow | Frost | Fog | Brit. | Contr. | Elastic | Pixel | JPEG | Avg. |
|---|---|---|---|---|---|---|---|---|---|---|---|---|---|---|---|---|
| Source | 10.7 | 11.7 | 11.3 | 22.1 | 14.6 | 23.5 | 21.2 | 30.4 | 29.1 | 33.6 | 51.8 | 16.1 | 12.4 | 29.5 | 30.8 | 23.3 |
| TENT [37] | 5.5 | 4.4 | 6.7 | 23.8 | 18.5 | 26.4 | 23.4 | 32.0 | 30.2 | 36.1 | 52.9 | 21.6 | 13.2 | 33.6 | 34.7 | 24.2 |
| ETA [25] | 18.1 | 19.2 | 19.7 | **25.1** | 21.7 | 28.7 | 25.7 | 34.5 | 31.7 | 38.0 | 53.9 | 25.8 | 17.1 | 36.3 | 36.7 | 28.8 |
| READ [43] | 15.4 | 14.4 | 12.4 | 21.8 | 14.0 | 23.2 | 20.9 | 30.5 | 29.3 | 33.7 | 52.0 | 15.8 | 12.3 | 29.5 | 30.7 | 23.7 |
| Ours | **20.6** | **22.3** | **20.9** | 24.5 | **25.0** | **30.5** | **27.7** | **35.4** | **32.3** | **40.4** | **54.5** | **30.2** | **26.4** | **38.2** | **38.6** | **31.2** |

# E  More Ablation Studies

**Variants on PDR.** In the original design of PDR, we take both prediction bias and confidence levels to formulate the reweighting indicator. In this subsection, we substitute softmax confidence scores of each data with their entropy. Specifically, we compute the quantile-based importance weighting using prediction bias and entropy levels instead of with softmax confidence scores. We show the results in Table 9. It is shown that softmax confidence scores are more effective in handling multi-modal domain shifts, which aligns with the findings of previous methods [50].

**Variants on AGA.** AGA is originally computed by MMD-based regularization on the attention maps. In this subsection, we substitute MMD-based regularization with MSE, L2, and KL divergence, and compare their performance in Table 11. Moreover, we change the attention maps to fusion features, fusion feature maps, and output logits, and summarize their performance in Table 12. The difference between fusion features and fusion feature maps lies in whether average pooling is applied. From Table 11, we observe that both element-wise and distribution-wise alignments underperforms MMD-based regularization, as the robust prior knowledge is not characterized by specific values or static patterns in the attention maps, but rather by dynamically changing relationships across modalities. From Table 12, we observe that the substitution options yield similar results, likely due to the prior knowledge across modalities are crucial for robust adaptation, whereas regularizing features and aligning logits appears to be ineffective.

# F  Theoretical Analysis

To strengthen the theoretical foundation of PDR, we provide a brief analysis of why prediction bias arises and how our design mitigates it. Suppose the output logits are $\mathbf{z}_i = [z_{i1}, \ldots, z_{iK}]$, we have the softmax probability $p_{ik} = \frac{e^{z_{ik}}}{\sum_j e^{z_{ij}}}$ and the entropy loss $\mathcal{L}_{\text{ent}} = -\frac{1}{N} \sum_{i=1}^{N} \sum_{k=1}^{K} p_{ik} \log p_{ik}$. We

first derive the gradient of the softmax function with respect to the logits: $\frac{\partial p_{ij}}{\partial z_{ik}} = p_{ij}\left(\delta_{jk} - p_{ik}\right)$, as well as the gradient of the entropy loss with respect to the softmax outputs: $\frac{\partial \mathcal{L}_{\text{ent}}}{\partial p_{ij}} = -\log p_{ij} - 1$. By applying the chain rule, we obtain the gradient of the entropy loss with respect to the logits as follows: $\frac{\partial \mathcal{L}_{\text{ent}}}{\partial z_{ik}} = \sum_{j=1}^{K} \frac{\partial \mathcal{L}_{\text{ent}}}{\partial p_{ij}} \cdot \frac{\partial p_{ij}}{\partial z_{ik}} = \sum_{j=1}^{K} \left(-\log p_{ij} - 1\right) \cdot p_{ij}\left(\delta_{jk} - p_{ik}\right)$, where the Kronecker delta: $\delta_{jk} = 1$ if $j = k$, else 0.

This expression shows that entropy minimization encourages the model to produce increasingly (over) confident (*i.e.*, low-entropy) predictions by amplifying the largest logit and suppressing the others, which is also discussed in previous works [26]. To address prediction bias in multi-modal domain shifts, PDR partitions each data batch into biased and unbiased subsets using a prediction bias indicator (Eq. (1)), which leverages batch-average prediction frequency to identify overconfident predictions, often false positives due to entropy minimization. These biased samples are regularized via entropy maximization (the second term in Eq. (2)), which counteracts noise propagation by encouraging uniform probability distributions, as supported by theoretical insights from [21, 17] on entropy regularization under noisy conditions.

Additionally, to balance bias and confidence levels and prevent issues like unreliable gradient from low-confidence predictions [25], we propose a quantile ranking strategy (Algorithm 2). This strategy reweights unbiased samples by assigning higher weights to confident predictions and lower weights to uncertain ones, ensuring stable and reliable test-time adaptation. The theoretical reasonability of this approach stems from its alignment with robust optimization principles, where reweighting mitigates the impact of noisy outliers, as discussed in [25]. We demonstrate PDR's consistent stability and superior performance across challenging multi-modal domain shift scenarios, including continual settings (Table 8) and changing environments (Section 4.4).

# G   Visualizations

**Attention map.** We visualize the attention maps of clean data, $\mathcal{A}^{+}$, and $\mathcal{A}^{-}$. It is shown that the distributions of "audio-audio" block and "video-audio" block in $\mathcal{A}^{+}$ are closer to those in clean data attention map compared to the distributions in $\mathcal{A}^{-}$. This demonstrates that AGA effectively focuses on robust information dependencies while suppressing the influence of sensitive ones.

**Confusion matrix.** We provide more visualization for confusion matrix in Figure 10. It is demonstrated that our method mitigates the false positive issue in the pre-trained model and generates more true positive predictions, which explains its effectiveness.

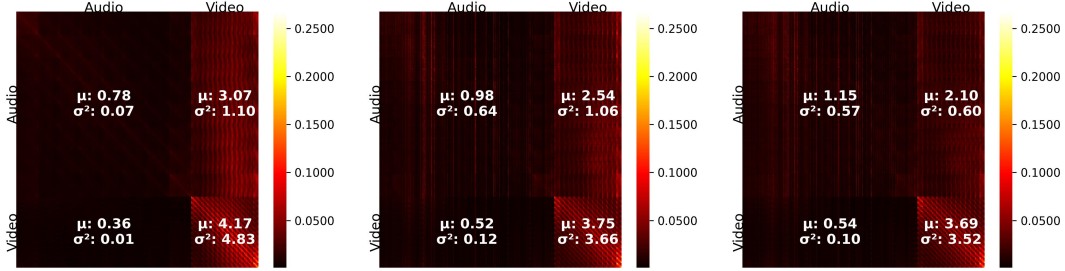

Figure 9: Visualization on the attention maps for clean data (left), $\mathcal{A}^{+}$ (middle), and $\mathcal{A}^{-}$ (right). The values are amplified by 10, 000 times for clarity. The number upon the blocks denotes the mean and variance. The corruption type for audio and video modalities are "Crowd" and "Defocus".

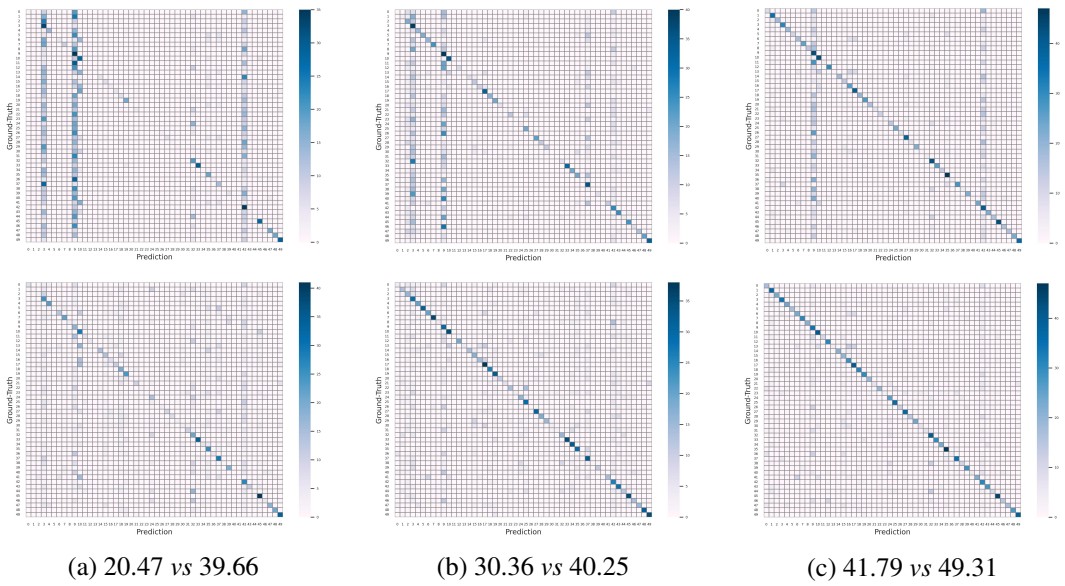

(a) 20.47 *vs* 39.66          (b) 30.36 *vs* 40.25          (c) 41.79 *vs* 49.31

Figure 10: Comparison on confusion matrix of predictions for the pre-trained model (Top) and our method (Bottom). The corruption type for audio and video modalities are "Crowd" and "Fog" in subfigure (a), "Crowd" and "Snow" in subfigure (b), and "Crowd" and "Zoom blur" in subfigure (c), respectively. "X *vs.* Y" denotes the performance comparison between the pre-trained model and PTA.

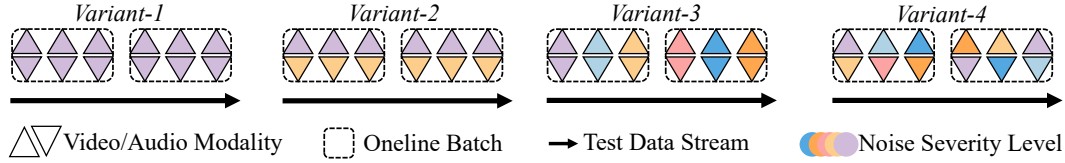

Figure 11: Illustration of multi-modal domain shifts with changing severity levels.

## H    Limitations

Test-time adaptation (TTA) focuses on the deployment phase of a pre-trained model, emphasizing the need for efficiency. Although our method achieves similar efficiency compared to existing methods, as demonstrated in Fig. 6, there is still room for improvement in efficiency for the sake of real-world deployment. Due to the limitation of computational resources, we only test our method on one GPU. However, it could be accelerated through parallel processing using multiple GPUs.

## I    Broader impacts

Test-time adaptation enables a model pretrained on source domain data to adapt to target domain data in real-time. It has broad applications in dynamic scenarios such as autonomous driving. To the best of our knowledge, our work does not have obvious negative social impacts.

