# OpenReview forum: "Partition-Then-Adapt: Combating Prediction Bias for Reliable Multi-Modal Test-Time Adaptation"
_NeurIPS.cc/2025/Conference — NeurIPS 2025 spotlight_

### Official Review · Reviewer_UQ6f · 2025-07-01

**Clarity:** 2
**Significance:** 1
**Originality:** 3
**Rating:** 4
**Confidence:** 3

**Summary:**

This paper tackles the under-explored problem of multi-modal test-time adaptation (MM-TTA) when all modalities simultaneously undergo domain shift. To mitigate prediction bias under this setting, the authors propose Partition-Then-Adapt (PTA),  which contains two key components: Partition and  Debiased Re-weighting (PDR) and Attention-Guided Alignment (AGA) . Extensive experiment show that PTA achieves good performance.

**Questions:**

Please refer to Weaknesses.

**Ethical Concerns:**

["NO or VERY MINOR ethics concerns only"]

**Final Justification:**

Based on the author's response and the comments of other reviewers, I gave the final score.

**Limitations:**

yes

**Paper Formatting Concerns:**

No significant formatting issues were observed.

**Quality:**

3

**Strengths And Weaknesses:**

Strengths:
1. This paper raises a meaningful question: how to design a test-time adaptation (TTA) scheme that mitigates domain shift when multiple modalities are involved. The authors observe that existing MM-TTA approaches focus only on single-modal domain shifts within multi-modal tasks; when all modalities are simultaneously shifted, performance degrades markedly. Hence, investigating how to develop an MM-TTA method for this more challenging setting is both timely and significant.

2. The paper is well organised,  clear writing, and figures/tables support the narrative effectively.

3. The authors benchmark PTA against MM-TTA methods, report clear SOTA gains, and conduct ablations on PDR, AGA, weighting strategies, and hyper-parameters, which help the reader understand each design choice.


Weaknesses:
1. Is “high class frequency ⇒ high bias” always valid? In PDR, samples whose predicted class appears more frequently than the batch mea  are deemed more biased. Yet class frequency can be high simply due to small batch size or genuine class imbalance in the target domain.  Please  provide results for PTA’s sensitivity to different batch sizes.

2. Possible misconception in the MMD explanation (§3.3)
    Minimising the squared MMD in Eq. (4) actually reducesthe distance between $A^+$ and $A^-$, i.e., makes the two distributions more similar.
   However, the text claims the goal is to “push them apart”. Could the authors reconcile this apparent contradiction or correct the description?

3. Definition of the confidence term $\mathcal K$
  Are they the maximum Softmax probabilities of the current classifier? Please add a precise definition.

---

> ### Author Rebuttal · Authors · 2025-07-30
>
> ### Response to weaknesses
>
> $\textbf{W1}$: Applicability to different batch sizes.
>
> $\textbf{A1}$: Thank you for your insightful comment. We agree this is an important question as online data is often insufficient, and at times, only a limited amount is available. And addressing this problem would firmly validate the applicability of our method dealing with real-world scenarios. In fact, we provide results for PTA’s applicability to different batch sizes in Fig. 8, Appendix D, Page 14, where small batch size has been also discussed. To handle extremely small batch sizes, we implement a memory queue to store historical predictions for computing Eq. (1) and (2), effectively adapting the original computation method for offline use. Therefore, PTA still works well and consistently outperform SOTAs in such cases.
>
> $\textbf{W2}$: The meaning of the third term in Eq. (4).
>
> $\textbf{A2}$: Thank you for your helpful review. We acknowledge the typo in Line 174 —  "penalizes" should in fact be "promotes". To avoid any misunderstanding, we will reframe Lines 174-175 as ''The third term regularizes the model by guiding $\mathcal{A}^-$ with $\mathcal{A}^+$. Specifically, it aligns the distributions of $\mathcal{A}^-$ with those of $\mathcal{A}^+$.''
>
> $\textbf{W3}$: Definition of the confidence term $\mathcal{K}$.
>
> $\textbf{A3}$: Thank you for your valuable feedback. We confirm that the confidence term ( $\mathcal{K}$ ) represents the maximum softmax probabilities of the current classifier for a given batch data. We will clarify this definition in Lines 133-134 as ''Let the prediction confidence of $\mathcal{X}$ be denoted as $\mathcal{K}(\mathcal{X}) = \max(\text{softmax}(\mathbf p(\mathcal{X})))$, which corresponds to the highest softmax probability produced by the current classifier for the input batch $\mathcal{X}$.''

---

> > ### Comment · Reviewer_UQ6f · 2025-08-06
> >
> > Thanks for the authors' response. It has addressed the majority of my concerns. I will maintain my original score.

---

> > > ### Author Response · Authors · 2025-08-06
> > >
> > > We appreciate your constructive comments and your recognition that our response has sufficiently addressed the main concerns raised. We will improve the paper accordingly based on your suggestions.

---

### Official Review · Reviewer_Jc1F · 2025-07-03

**Clarity:** 3
**Significance:** 3
**Originality:** 2
**Rating:** 5
**Confidence:** 4

**Summary:**

In this paper, the authors address the problem of test-time adaptation (TTA) under multi-modal domain shifts, specifically in the context of video and audio co-classification using pre-trained video and audio encoders. While prior TTA methods for multi-modal settings primarily focused on single-modality domain shifts, the authors propose a novel method, called Partition-Then-Adapt (PTA), to tackle domain shifts in both modalities simultaneously.

PTA consists of two key components:
- Partition and Debiased Reweighting (PDR), which identifies reliable and unreliable samples in a test batch by quantifying prediction bias—measured as the frequency of each predicted label relative to the batch average. Samples with lower bias are considered reliable, while those with higher bias are considered unreliable. A quantile-based reweighting strategy is applied to focused on reliable samples and remove effect of unreliable ones.
- Attention-Guided Alignment (AGA), which aligns the attention maps of unreliable samples with those of reliable ones by minimizing the maximum mean discrepancy (MMD) between their attention distributions.

Experiments on Kinetics50-C and VGGSound-C demonstrate the effectiveness of the proposed method, showing consistent improvements over existing TTA approaches.

**Questions:**

- Why does the SuMi method fail to adapt the pre-trained model stably across all benchmarks?
According to Tables 6 and 7, SuMi not only performs inconsistently under multi-modal domain shifts but also fails to improve performance under single-modal shifts. An in-depth analysis of SuMi's failure modes would benefit readers.

- Why do multi-modal domain shifts degrade model calibration more severely than single-modal shifts, as shown in Figure 1(c)?
This observation is central to the motivation of PTA but is not sufficiently explained. Additional insights into why the interaction between modalities causes such a pronounced miscalibration—possibly due to conflicting noise patterns or misaligned feature distributions—would strengthen the justification for the proposed method.

- Why is the proposed method still effective under single-domain shifts?
While PTA is designed for multi-modal shifts, the results suggest that it remains effective in single-modality scenarios. A deeper explanation is warranted: for instance, does prediction bias still emerge due to label imbalance or temporal inconsistencies even in single-domain cases? Clarifying this would demonstrate the generality of the framework.

- Can the proposed method outperform existing single-modality TTA methods on standard benchmarks such as CIFAR-C or ImageNet-C?
If so, a comparison table on these datasets would help illustrate the broader applicability of PDR and AGA, even beyond multi-modal settings. However, due to time constraints, this is a minor suggestion.

- Can the proposed method also be applied to other multi-modal classification models?

**Ethical Concerns:**

["NO or VERY MINOR ethics concerns only"]

**Final Justification:**

The authors' rebuttal has adequately addressed the major concerns raised in the initial review. Specifically, they demonstrate that PTA performs well across different architectures, including CLIP. Additionally, they show that PTA is robust to small batch sizes, which is important for real-world applications. Based on these clarifications and results, I am raising my initial rating to accept.

**Limitations:**

Authors discuss the limitations in Appendix. However, I encouraged authors to fix the following points to improve the quality of the paper.

1. I suggest the authors include the efficiency of the Source (i.e., the original pre-trained model without adaptation) in Figure 6. Since TENT involves only a few trainable parameters, I assume the Source would be slightly faster. However, readers may want a direct comparison of the efficiency (e.g., latency or parameter count) between the Source model and various TTA methods. This is a minor suggestion for improving clarity.

2. Additionally, I recommend that the authors explicitly specify the unit for the number of trainable parameters shown in Figure 6. I assume it is in millions, but making this clear in the figure caption or axis label would avoid confusion. Moreover, in Tables 6 and 7, the authors only highlight (bold) their own method's results. To provide a fair comparison and improve readability, it would be better to bold the best values across all methods, not just the proposed one.

**Paper Formatting Concerns:**

I have no concerns on paper formatting.

**Quality:**

3

**Strengths And Weaknesses:**

### Strengths
- The paper is well-structured and easy to follow.
- The authors address the problem of multi-modal domain shifts, which has been underexplored in the literature.
- The proposed method is well-motivated, and the Attention-Guided Alignment (AGA) component effectively mitigates the issue of uniform attention maps that can arise when PDR is used for adaptation.

### Weaknesses
- I believe that the method may require a large batch size to accurately estimate prediction bias and perform effective partitioning, which could limit its applicability in low-latency or memory-constrained environments.
- The key assumption—that multi-modal domain shifts lead to performance drops even for high-confidence predictions (as shown in Figure 1(c))—is only validated using CAV-MAE. The generality of this observation across other model architectures is not validated.

---

> ### Author Rebuttal · Authors · 2025-07-30
>
> ### Response to weaknesses
>
> $\textbf{W1}$: Applicability to different batch sizes.
>
> $\textbf{A1}$: Thank you for raising this important point. We agree that small batch sizes are common in real-world, low-latency scenarios and must be addressed. To this end, we evaluate our method under various batch sizes in Fig. 8 (Appendix D, Page 14). Results show that PTA remains robust and consistently outperforms SOTAs even with small batches. To mitigate bias estimation instability, we introduce a memory queue that stores historical predictions for computing Eq. (1) and (2). This allows more stable frequency estimation without requiring large batch sizes. This design ensures that PTA remains practical and effective under constrained settings.
>
> $\textbf{W2}$: Clarification of some assumptions.
>
> $\textbf{A2}$: Fig. 3 (c) presents an example illustrating that high-confidence predictions can become unreliable under multi-modal domain shifts. We adopt CAV-MAE in Fig. 3 (c) to ensure a fair comparison, as it is also used by previous methods [9, 31, 38]. The results clearly show that the positive rate of high-confidence predictions drops significantly under such domain shifts. Moreover, the empirical results reveals that addressing this prediction bias issue substantially improves adaptation performance  (see Tables 1-3). For results obtained with other model architectures, please kindly refer to Q/A5. Additional figures validating this observation with other model architectures will be included in the Appendix.
>
> ### Response to questions
>
> $\textbf{Q1}$: In-depth analysis of SuMi.
>
> $\textbf{A1}$: Thank you for your constructive feedback. For fair comparison, we reimplement all comparison methods using their official code (if any) and report the average accuracy across 5 independent seeds under both single- and multi-modal shifts. SuMi outperforms the pre-trained model (Source) under single-modal domain shifts (61.01 vs. 60.58 in Table 6, and 69.88 vs. 69.32 in Table 7), but fails under multi-modal domain shifts (see Tables 1–2). The reason might originates from SuMi’s design. Specifically, the key components of SuMi are (1) uni-modal assistance (UA) with entropy-based reweighing (EW) and (2) KL-based mutual information sharing.
>
> First, as we demonstrate in the empirical results (Lines 239-248) and the analysis (Lines  249-260 and Fig. 3 (b)), UA+EW assigns high weights to the selected candidates, regardless of whether they are false positives or not, which may poses error accumulation. Second, the KL-based loss function may not provide meaningful mutual information under severe multi-modal domain shifts, as significantly performance drop of the pre-trained model (Fig. 1 (a)) occurs. As a result, the above properties could be the reason why SuMi achieves unsatisfactory performance for MMTTA. We will add the above discussion in Section 4.
>
> $\textbf{Q2}$: Degradation of the pre-trained model.
>
> $\textbf{A2}$: Thank you for your insightful suggestion. We agree with your hypothesis that multi-modal domain shifts degrade model calibration more severely than single-modal shifts, likely due to conflicting noise patterns or misaligned feature distributions across modalities. Multi-modal domain shifts degrade model calibration primarily because they induce complex interactions between modalities in the fusion block, leading to degraded discrimination of fused features (Fig. 1 (a)). These skewed feature representations bias the classifier toward a few dominant classes (Fig. 1 (b)). Consequently, the normalized entropy, computed from these biased predictions, significantly impairs model calibration (Fig. 1 (c)). We will improve the presentation in Lines 31-34 with the above discussion.
>
> $\textbf{Q3}$: Explain the effectiveness of PTA under single-domain shfits.
>
> $\textbf{A3}$: Thank you for your thoughtful comment. We agree that PTA effectively addresses prediction bias in single-modal domain shifts, though the issue is less pronounced than in multi-modal shifts. First, our experiments show that the pre-trained model’s performance degrades under both single- and multi-modal domain shifts (Fig. 1 (a)), leading to increasingly noisy (incorrect) predictions. Entropy minimization exacerbates this by sharpening low-entropy predictions, regardless of their reliability, which propagates noise and risks a collapsed trivial solution. Empirical results from Tent (Tables 1–2 and Table 6) confirm that excessive entropy minimization with noisy predictions can degrade performance compared to the pre-trained model (Source). Second, following prior works [9, 31, 38], we use randomly sampled batch data, ensuring no intentional label imbalance. This indicates that prediction bias primarily arises from noisy predictions, further amplified by entropy minimization, rather than imbalanced data.
>
> $\textbf{Q4}$: Add experiments on CIFAR10/100-C and ImageNet-C.
>
> $\textbf{A4}$: Thanks for your suggestion. We provide more experiments on CIFAR-10/100-C and ImageNet-C and report the average accuracy across 15 types of corruptions (severity level 5).
>
> | Method | CIFAR10-C | CIFAR100-C | ImageNet-C  |
> |--------|-------|--------|-------|
> | Source | 56.5 | 53.6  | 18.0 |
> | TENT   | 81.7 | 68.5  | 34.7 |
> | ETA    | **81.9** | **68.9**  | 39.7 |
> | READ   | 79.0 | 61.5  | 31.3 |
> | Ours   | 81.3 | 68.8 | **41.3** |
>
> We observe that our method outperforms READ on single-modal datasets and matches specialized single-modal methods (surpassing on IN-C). Crucially, it excels over all SOTAs in challenging multi-modal domain shifts (see Tables 1–2). We will add the comparison table on these datasets to help illustrate the broader applicability of our method following your advice.
>
> $\textbf{Q5}$: Performance on other multi-modal classification models.
>
> $\textbf{A5}$: Thank you for your valuable feedback. First, we validate our method’s effectiveness on synthetic multi-modal domain shifts (see Tables 1-2 and Tables 4-7) using CAV-MAE [8, 31] as the backbone. Second, we demonstrate our method's generalization to real-world domain shifts using a Transformer-based multi-modal classification model [10] as the backbone (see Table 3). Third, we prove our method's broader applicability using CLIP (ViT-B-16) as the backbone. To simulate multi-modal domain shifts, we use ImageNet-C for the CLIP's image branch and apply Gaussian Noise to its text branch.  We report the comparison table below:
>
> |  | Gauss. | Shot  | Impul. | Defoc. | Glass | Motion | Zoom  | Snow  | Frost | Fog   | Brit. | Contr. | Elastic | Pixel | JPEG  | AVG   |
> |--------|--------|-------|--------|--------|-------|--------|-------|-------|-------|-------|-------|--------|---------|-------|-------|-------|
> | Source | 10.7   | 11.7  | 11.3   | 22.1   | 14.6  | 23.5   | 21.2  | 30.4  | 29.1  | 33.6  | 51.8  | 16.1   | 12.4    | 29.5  | 30.8  | 23.3  |
> | TENT   | 5.5    | 4.4   | 6.7    | 23.8   | 18.5  | 26.4   | 23.4  | 32.0  | 30.2  | 36.1  | 52.9  | 21.6   | 13.2    | 33.6  | 34.7  | 24.2  |
> | ETA    | 18.1   | 19.2  | 19.7   | **25.1**   | 21.7  | 28.7   | 25.7  | 34.5  | 31.7  | 38.0  | 53.9  | 25.8   | 17.1    | 36.3  | 36.7  | 28.8  |
> | READ   | 15.4   | 14.4  | 12.4   | 21.8   | 14.0  | 23.2   | 20.9  | 30.5  | 29.3  | 33.7  | 52.0  | 15.8   | 12.3    | 29.5  | 30.7  | 23.7  |
> | Ours   | **20.6**   | **22.3**  | **20.9**   | 24.5   | **25.0**  | **30.5**   | **27.7**  | **35.4**  | **32.3**  | **40.4**  | **54.5**  | **30.2**   | **26.4**    | **38.2**  | **38.6**  | **31.2**  |
>
> In summary, out method demonstrate its robustness on multiple multi-modal classification models.
>
> ### Response to limitations
>
> $\textbf{L1}$: Improve Fig. 6 by including the efficiency of the Source.
>
> $\textbf{A1}$: We appreciate your feedback. The number of samples processed per second (samples/sec) by Source is 92.4.
> We will revise Fig. 6 by including the Source in the comparison to improve clarity, following your advice.
>
> $\textbf{L2}$: Improve Fig. 6, and Tables 6-7.
>
> $\textbf{A2}$:  We appreciate your valuable suggestions. The unit for the number of trainable parameters in Fig. 6 is indeed millions (M). We will revise Fig. 6 by adding the unit to the y-axis for the number of trainable parameters and improve Tables 6-7 by highlighting the best-performence of comparison methods, following your advice.

---

> > ### Comment · Reviewer_Jc1F · 2025-08-05
> >
> > I thank the authors for conducting the additional experiments and providing a detailed response to my review. The major concerns have been adequately addressed. I suggest that the authors include the experiments with CLIP in the main paper to demonstrate the generalization capability of PTA.

---

> > > ### Author Response · Authors · 2025-08-05
> > >
> > > We sincerely thank you for the positive feedback and valuable suggestions. We will include the experiments with CLIP in the main paper to further highlight the generalization ability of PTA, as recommended.

---

### Official Review · Reviewer_Y3nk · 2025-07-03

**Clarity:** 3
**Significance:** 3
**Originality:** 3
**Rating:** 5
**Confidence:** 3

**Summary:**

This paper highlights the limitations of using confidence or entropy-based measures for selective test-time adaptation, particularly in multi-modal domains. To address this, the authors propose a novel Partition then Adapt (PTA) strategy, which first clusters data before applying adaptation, aiming to improve robustness and reliability.

**Questions:**

Please see weaknesses and questions section.

**Ethical Concerns:**

["NO or VERY MINOR ethics concerns only"]

**Final Justification:**

The authors addressed the raised concern. I am keeping my score.

**Limitations:**

yes

**Quality:**

3

**Strengths And Weaknesses:**

**Strengths**
- The paper addresses an important and practical issue that can arise in multi-modal domains, where traditional confidence-based measures may fail.
- The proposed method leverages attention mechanisms effectively to improve adaptation.
- The ablation study is thorough and helps clarify the contribution of each component.
- Extensive experiments across multiple datasets support the efficacy of the proposed approach.

**Weaknesses and Questions**

- The proposed method, PTA, quantifies prediction bias based on prediction frequency. Specifically, according to Eq. (1), if a sample's predicted label has a higher frequency than the average frequency of predicted labels within the batch, the sample is considered a biased (i.e., unreliable) one. While the empirical results based on this partitioning are convincing, it would be helpful to include a higher-level explanation or intuition for why partitioning reliable and unreliable samples based on prediction frequency should work. Could the authors provide supporting evidence or cite prior work that inspired this approach?

---

> ### Author Rebuttal · Authors · 2025-07-30
>
> ### Response to weaknesses and questions
>
> $\textbf{WQ1}$: Supporting evidence.
>
> $\textbf{A1}$: Thank you for the valuable suggestion. The intuition behind our use of prediction frequency stems from the well-documented effect of entropy minimization under domain shifts: it tends to amplify spurious confident predictions, often leading to a collapsed solution where only a few classes dominate [20]. This is especially problematic in multi-modal settings, where noisy modalities can further reinforce such bias.
> To address this, we quantify prediction bias using the frequency of predicted labels within each batch. Samples assigned to dominant classes, i.e., those predicted more frequently than the batch average, are likely influenced by such bias.
> Fig. 3 (c) and (d) demonstrate that our design effectively mitigates prediction bias in multi-modal domain shifts, significantly reducing false positive rates for the two most biased classes (24.2 $\rightarrow$ 5.7 and 24.0 $\rightarrow$ 6.7).
> Moreover, our empirical results in Fig. 5 (e) confirm that using batch-average frequency as the bias threshold outperforms static baselines.
> To the best of our knowledge, we are the first to propose prediction frequency as a bias indicator for partitioning reliable/unreliable samples in the MMTTA setting. We will add this discussion and cite relevant works in the final version.

---

> > ### Comment · Reviewer_Y3nk · 2025-08-05
> >
> > Thank you for the author's reply. Please add regarding discussion and cite relevant works as mentioned in the final version of the paper.

---

> > > ### Author Response · Authors · 2025-08-05
> > >
> > > We appreciate your constructive feedback. We will add the related discussion and include citations to the relevant works in the final version of the paper,  as suggested.

---

### Official Review · Reviewer_xcN3 · 2025-07-03

**Clarity:** 1
**Significance:** 2
**Originality:** 2
**Rating:** 4
**Confidence:** 2

**Summary:**

This paper introduces PTA for test-time adaptation under multi-modal domain shifts. The method contains two components, Partition and Debiased Reweighting (PDR) and multi-modal Attention-Guided Alignment (AGA). PDR splits the test batch into reliable and unreliable subsets using a bias indicator, and then employs quantile-based strategy to jointly consider prediction bias and confidence to reweight samples. AGA adds an MMD-based regularization loss on attention maps.

**Questions:**

See Weakness

**Ethical Concerns:**

["NO or VERY MINOR ethics concerns only"]

**Final Justification:**

I appreciate the authors’ response. I have read the rebuttal carefully, and it addresses most of my concerns.

One issue remains—W5. The single-modal domain-shift results in Tables 6 and 7 differ substantially from those reported in three prior papers (ABPEM [38], SuMi [9], and READ [31]), yet the manuscript does not provide sufficient implementation details to account for these gaps.

In the rebuttal, the authors suggest that “minor discrepancies … may arise due to differences in GPUs and random seeds,” but the average accuracies for ABPEM (65.0 vs. 63.34) and SuMi (63.9 vs. 61.01) do not seem to be “minor.”

Because I am not deeply familiar with the implementation specifics of the previous methods—and because the majority of my other concerns have been resolved, I have raised my overall score to 4 and reduced my confidence score to 2. I expect the authors to include a more detailed discussion of how previous baselines were re-implemented in the revised manuscript, which can better benefit the community.

[38] Yusheng Zhao, Junyu Luo, Xiao Luo, Jinsheng Huang, Jingyang Yuan, Zhiping Xiao, and Ming Zhang. Attention bootstrapping for multi-modal test-time adaptation. In AAAI, 2025.

[9] Zirun Guo and Tao Jin. Smoothing the shift: Towards stable test-time adaptation under complex multimodal noises. In ICLR, 2025.

[31] Mouxing Yang, Yunfan Li, Changqing Zhang, Peng Hu, and Xi Peng. Test-time adaptation against multi-modal reliability bias. In ICLR, 2024.

**Limitations:**

yes

**Quality:**

3

**Strengths And Weaknesses:**

## Strength

- This paper is well written and follows a good structure.
- Promising results are obtained in the studied benchmarks.
- The effectiveness of the proposed two components is also shown.

## Weakness

- Some claims might not be convincing. For example, the authors claim “Fig. 1 illustrate the effectiveness of existing reweighting strategies diminishes under multi-modal domain shifts” in Line 116. However, Fig. 1 only evaluates the pretrained model, and dose not provide direct evidence or analysis for current methods (e.g., SuMi, READ) in multi-modal domain shifts. The logical connection about the motivation part is weak and requires more elaboration to be convincing.
- The authors experimentally demonstrate the effectiveness of proposed sample reweighting strategy. However, the core idea of splitting samples into reliable/unreliable subsets based on confidence/bias indicators and give unreliable samples little or negative weights has been introduced by earlier methods (e.g. DeYO[1]). The authors are encouraged to conduct a deeper comparison and discuss more about how PDR differs from, or improves upon those prior reweighting strategies.
- Some necessary citations might be missing and related work might not be sufficient. For instance, using weighted entropy loss in TTA (Line 152) has already been proposed in TTL[2].
- Key experimental details are missing. (1) What does the “source model” denote? (2) There lacks essential details about CAV-MAE architecture, such as the number of attention layers used for fusion. (3) It is not clear how many attention layers are updated during adaption and what fraction of total parameters they represent. (4) In Table1, labeling the video+audio modality corruptions as Kinetics50-C may cause confusion with the single-modality Kinetics50-C benchmark. (5) It is not entirely clear what each number in Table 1 and 2 means? (e.g., average over six audio corruptions in Table 1?)
- Baseline reproduction discrepancies. It's not clear why results of SOTA methods (e.g., SuMi, READ) differ from their original papers in Table 6 and Table 7.
- Concerns about hyper-parameter fairness for baseline comparison. Figure 5 shows that learning rate and updated modules greatly influence TTA performance, yet PTA and baselines use different learning rates. It is encouraged to supplement comparisons with different learning rates and updating modules, which would make the comparisons more comprehensive.
- Although the experiments demonstrate the method’s effectiveness, it remains unclear whether PTA can generalize to other single-modal and multimodal datasets (e.g, image-text). It is encouraged to conduct experiments to verify the method’s applicability to vision-language models (e.g., CLIP in TDA[5]) and image shift contexts (e.g., ImageNet-C in FOA[3],  CIFAR10-C in SoTTA[4]). Such experiments would validate PTA’s broader applicability.

[1] Entropy is not enough for test-time adaptation: From the perspective of disentangled factors. In ICLR, 2024

[2] Test-Time Low Rank Adaptation via Confidence Maximization for Zero-Shot Generalization of Vision-Language Models. In WACV, 2025

[3] Test-Time Model Adaptation with Only Forward Passes. In ICML, 2024

[4] Robust Test-Time Adaptation on Noisy Data Streams In NeurIPS, 2023

[5] Efficient Test-Time Adaptation of Vision-Language Models. In CVPR, 2024

---

> ### Author Rebuttal · Authors · 2025-07-30
>
> ### Response to weaknesses
>
> $\textbf{W1}$: Evidence of some claims.
>
> $\textbf{A1}$: Thanks for your valuable feedback. The empirical results in Tables 1-3 and 6-7 provide direct evidence that existing methods perform less effectively under multi-modal domain shifts compared to single-modal domain shifts. For example, compared to the pre-trained model, READ achieves average performance gains of 2.7% across two independent single-modal shift scenarios and 3.6% on multi-modal domain shifts.  In contrast, our method demonstrates superior gains of 3.1% (single-modal) and 9.7% (multi-modal) respectively, showing significantly stronger robustness.
> Moreover, we have provided a deeper analysis in Fig. 3 (a), which illustrates that existing reweighting strategies, such as UA+EW and LE+RA, are less effective under multi-modal domain shifts compared to our method. To strengthen the logical connection of the context, we will reframe Line 116 as "However, as we demonstrate in the experimentation section (Tables 1-3 and 6-7), the effectiveness of existing reweighting strategies become less effective under multi-modal domain shifts compared to single-modal domain shifts."
>
>
> $\textbf{W2}$: Comparison with DeYO.
>
> $\textbf{A2}$: Thank you for pointing this out. We have already included a comparison and discussion of representative reweighting methods in Fig. 3 (a) and Lines 239–265. Specifically, while DeYO [1] introduces a selective reweighting strategy, it operates on a strict subset of low-entropy samples with high PLPD scores (see Eqs. 8–10 in [1] and its official implementation). This design overlooks the accumulation of prediction bias that is especially critical in multi-modal domain shifts, where confident but biased predictions are common. In contrast, our PDR module first quantifies batch-level prediction bias (Eq. (1)) to partition the data into potentially reliable and unreliable subsets, and then reweights them accordingly (Eq. (2)). This allows PDR to explicitly suppress false positives and promote reliable adaptation beyond confidence alone. Moreover, we have conducted a new experiment under multi-modal domain shift on Kinetics50 using the same learning rate and update modules. Our method achieves 41.5% accuracy, significantly outperforming DeYO’s 28.5%. We will incorporate these results and discussions into Section 4 and add DeYO to Fig. 3 (a) and (b) for a more comprehensive comparison.
>
> $\textbf{W3}$: More citations.
>
> $\textbf{A3}$: Thank you for the helpful suggestion. We acknowledge that TTL [2] also employs weighted entropy loss. However, the reweighting strategies differ: TTL focuses on vision-language tasks by reweighting original and augmented views within the same sample, while our method operates on multi-modal domain shifts by partitioning the batch based on prediction bias and confidence. Unlike TTL and ETA, which primarily rely on entropy, our PDR module explicitly identifies and suppresses biased predictions, a key issue under multi-modal shifts. We will cite TTL and include this distinction in the related work section to clarify our contribution.
>
> $\textbf{W4}$: Experimental details.
>
> We sincerely value your detailed comments.
>
> $\textbf{W4.1}$: The ''Source model'' denotes the CAV-MAE pre-trained model (see Lines 198-199) for experiments on synthetic multi-modal domain shifts, and denotes a distinct transformer-based model [10] (see Lines 201-202) for the experiments on real-world domain shifts. The ''Source'' represents the performance of the pre-trained model for direct inference without any adaptation. We will add the meaning of ''Source'' in Line 196 and Appendix C.
>
> $\textbf{W4.2}$: We adopt CAV-MAE as the backbone architecture following [9, 31, 38]. Specifically, CAV-MAE employs an encoder-decoder architecture pre-trained on large-scale video datasets using both contrastive learning and masked image modeling. Its encoders both comprise 11 modality-specific Transformer layers for feature extraction, followed by an additional layer for cross-modal fusion. For more details, please kindly refer to [8, 31]. We will incorporate the details of CAV-MAE in Appendix B.
>
> $\textbf{W4.3}$: As we declare in Lines 204-205, ''Following [31, 38], we update query/key/value transformation matrices of the attention layer in the fusion block''. To be more specific, the total parameters are 1.8M (1% of total), which remain the same as [31, 38].
>
> $\textbf{W4.4}$: To eliminate the confusion, we will change Kinetics50 with multi-modal domain shifts as Kinetics50-MC (multi-modal corruption) and remain Kinetics50 with single-modal domain shifts as Kinetics50-C following your advice.
>
> $\textbf{W4.5}$: In Tables 1-2, each number denotes the average performance over six audio corruptions under a specific video corruption. We will add the meaning of each number to the captions of Tables 1-2.
>  We also present detailed results regarding the performance under a specific audio corruption paired with a specific video corruption in Table 10 of the Appendix.
>
> $\textbf{W5}$: Reproduction of comparison methods.
>
> $\textbf{A5}$: Thank you for the observation. To ensure fair comparison, we reimplemented all methods using their official code (if available), under the same adaptation protocol [31] using their default hyper-parameter settings. We report the average over 5 random seeds. Minor discrepancies with the original papers may arise due to differences in  GPUs and random seeds.
> Even when compared to SuMi’s original results, our method outperforms it under video-modal shifts (64.7 vs. 63.9) and achieves comparable performance under audio-modal shifts (71.4 vs. 71.9). We guarantee that all methods are implemented under fair conditions.
> We will clarify these implementation details and hyperparameter settings in Appendix C to ensure transparency and reproducibility.
>
> $\textbf{W6}$: Hyper-parameters.
>
> $\textbf{A6}$: Thank you for your insightful review. We use the baseline READ's default learning rate of 1e-4. As Fig. 5 (c) and (d) already show our method's performance with varying learning rates and updating modules, we assume you seek a comparison with other methods under identical settings. For clarity, we provide the following tables:
> |     LR    | 0.0001 | 0.0002 | 0.0003 | 0.0005 |
> |---------|--------|--------|--------|--------|
> | READ    | 35.4   | 35.8   | 36.0   | 34.8   |
> | Ours    | **40.8**   | **41.5**   | **40.9**   | **40.6**   |
>
> |    Params     | W\_QKV | W\_LN         | W\_WKV&LN      |
> |---------|--------|---------------|----------------|
> | READ    | 35.8   | Out of Memory | Out of Memory  |
> | Ours    | **41.5**   | **39.2**          | **41.8**           |
>
> We have the following observations: (1) our method consistently outperforms READ under the same learning rates and updating modules; (2) our method is memory-efficient, requiring significantly less GPU memory compared to READ.
>
> $\textbf{W7}$: Add experiments on CIFAR-C and ImageNet-C.
>
> $\textbf{A7}$: Thank you for your constructive and thoughtful feedback. Follow your advice, we run experiments on single-modal datasets, such as CIFAR10/100-C and ImgeNet-C using CNN-based backbones. We report the average accuracy over 15 types of corruptions for  these benchmarks in the following table:
>
> | Method | CIFAR10-C | CIFAR100-C | ImageNet-C  |
> |--------|-------|--------|-------|
> | Source | 56.5 | 53.6  | 18.0 |
> | TENT   | 81.7 | 68.5  | 34.7 |
> | ETA    | **81.9** | **68.9**  | 39.7 |
> | READ   | 79.0 | 61.5  | 31.3 |
> | Ours   | 81.3 | 68.8 | **41.3** |
>
> We observe that our method outperforms READ on single-modal datasets and matches specialized single-modal methods (surpassing on ImageNet-C). Crucially, it excels over all SOTAs in challenging multi-modal domain shifts (see Tables 1-2).
>
> We also run experiments on CLIP (ViT-B-16). To simulate multi-modal domain shifts, we use ImageNet-C for the image branch and apply Gaussian Noise to the text branch. For your convenience, we attach the performance below:
>
> |  | Gauss. | Shot  | Impul. | Defoc. | Glass | Motion | Zoom  | Snow  | Frost | Fog   | Brit. | Contr. | Elastic | Pixel | JPEG  | AVG   |
> |--------|--------|-------|--------|--------|-------|--------|-------|-------|-------|-------|-------|--------|---------|-------|-------|-------|
> | Source | 10.7   | 11.7  | 11.3   | 22.1   | 14.6  | 23.5   | 21.2  | 30.4  | 29.1  | 33.6  | 51.8  | 16.1   | 12.4    | 29.5  | 30.8  | 23.3  |
> | TENT   | 5.5    | 4.4   | 6.7    | 23.8   | 18.5  | 26.4   | 23.4  | 32.0  | 30.2  | 36.1  | 52.9  | 21.6   | 13.2    | 33.6  | 34.7  | 24.2  |
> | ETA    | 18.1   | 19.2  | 19.7   | **25.1**   | 21.7  | 28.7   | 25.7  | 34.5  | 31.7  | 38.0  | 53.9  | 25.8   | 17.1    | 36.3  | 36.7  | 28.8  |
> | READ   | 15.4   | 14.4  | 12.4   | 21.8   | 14.0  | 23.2   | 20.9  | 30.5  | 29.3  | 33.7  | 52.0  | 15.8   | 12.3    | 29.5  | 30.7  | 23.7  |
> | Ours   | **20.6**   | **22.3**  | **20.9**   | 24.5   | **25.0**  | **30.5**   | **27.7**  | **35.4**  | **32.3**  | **40.4**  | **54.5**  | **30.2**   | **26.4**    | **38.2**  | **38.6**  | **31.2**  |
>
> In summary, these additional experiments further demonstrate a broader applicability of our method.

---

> > ### Comment · Reviewer_xcN3 · 2025-08-05
> >
> > I appreciate the authors’ response. I have read the rebuttal carefully, and it addresses most of my concerns.
> >
> > One issue remains—W5. The single-modal domain-shift results in Tables 6 and 7 differ substantially from those reported in three prior papers (ABPEM [38], SuMi [9], and READ [31]), yet the manuscript does not provide sufficient implementation details to account for these gaps.
> >
> > In the rebuttal, the authors suggest that “minor discrepancies … may arise due to differences in GPUs and random seeds,” but the average accuracies for ABPEM (65.0 vs. 63.34) and SuMi (63.9 vs. 61.01) do not seem to be  “minor.”
> >
> > Because I am not deeply familiar with the implementation specifics of the previous methods—and because the majority of my other concerns have been resolved, I have raised my overall score to 4 and reduced my confidence score to 2. I expect the authors to include a more detailed discussion of how previous baselines were re-implemented in the revised manuscript, which can better benefit the community.
> >
> > [38] Yusheng Zhao, Junyu Luo, Xiao Luo, Jinsheng Huang, Jingyang Yuan, Zhiping Xiao, and Ming Zhang. Attention bootstrapping for multi-modal test-time adaptation. In AAAI, 2025.
> >
> > [9] Zirun Guo and Tao Jin. Smoothing the shift: Towards stable test-time adaptation under complex multimodal noises. In ICLR, 2025.
> >
> > [31] Mouxing Yang, Yunfan Li, Changqing Zhang, Peng Hu, and Xi Peng. Test-time adaptation against multi-modal reliability bias. In ICLR, 2024.

---

> > > ### Author Response · Authors · 2025-08-05
> > >
> > > We are grateful that you acknowledged our efforts in addressing almost all of your concerns and accordingly raised your score. To promote transparency and reproducibility, we will release the source code. Additionally, we will include a more detailed discussion of the re-implementation of previous methods in the Appendix, as you suggested.

---

### Official Review · Reviewer_h6TZ · 2025-07-05

**Clarity:** 3
**Significance:** 3
**Originality:** 3
**Rating:** 4
**Confidence:** 4

**Summary:**

This paper investigates the test-time adaptation problem in multi-modal scenarios where both single-modality and multi-modality domain shifts may exist. The authors propose an approach dubbed Partition-Then-Adapt (PTA) to address this new setting. PTA consists of two main components, i.e., Partition and Debiased Reweighting (PDR) and multi-modal Attention-Guided Alignment (AGA). PDR leverages the relative prediction frequency to split testing samples into reliable and unreliable subsets and adopts contrary adaptive weighting for the entropy minimization of the two sets. AGA can further regularize the adaptation of unreliable sets to focus on semantically meaningful multi-modal cues. Extensive empirical results demonstrate PTA consistently outperforms state-of-the-art TTA methods on multi-modal tasks.

**Questions:**

See weaknesses.

**Ethical Concerns:**

["NO or VERY MINOR ethics concerns only"]

**Final Justification:**

I intend to retain my initial rating for this submission.

**Limitations:**

Yes.

**Quality:**

3

**Strengths And Weaknesses:**

## Strengths
- The problem of test-time adaptation (TTA) is practical and significant. Moreover, according to the authors, the submission is the first to study multi-modal TTA with multiple domain shifts instead of single-modality shifts.

- The proposed PTA method is novel, especially the AGA component, which specifically addresses the challenge of multi-modal adaptation with unreliable samples.

- Experiments and ablation studies are very comprehensive. Extensive experiments demonstrate the superior performance of PTA over other TTA methods.

## Weaknesses
- The PDR component lacks solid support, such as a theoretical analysis of its reasonability.
- The presentation lacks clarity. The introduction of PDR is confusing due to imprecise notation. The specific meanings of $Z$ and $K$ are unclear—for example, it’s not specified whether they refer to scalars or vectors, or to individual samples or mini-batches.

---

> ### Author Rebuttal · Authors · 2025-07-30
>
> ### Response to weaknesses
>
> $\textbf{W1}$: Theoretical analysis.
>
> $\textbf{A1}$：Thanks for your suggestion. To strengthen the theoretical foundation of PDR, we provide a brief analysis of why prediction bias arises and how our design mitigates it.
>
> Suppose the output logits are $\mathbf{z}\_i = [z_{i1}, \dots, z_{iK}]$, we have the softmax probability $p\_{ik} = \frac{e^{z\_{ik}}}{\sum\_{j} e^{z\_{ij}}}$ and the entropy loss $\mathcal{L}\_{\mathrm{ent}} = -\frac{1}{N} \sum\_{i=1}^N \sum\_{k=1}^K p_{ik} \log p\_{ik}$.
>
> We first derive the gradient of the softmax function with respect to the logits: $\frac{\partial p_{i j}}{\partial z_{i k}}=p_{i j}\left(\delta_{j k}-p_{i k}\right)$,
> as well as the gradient of the entropy loss with respect to the softmax outputs: $\frac{\partial \mathcal{L}\_{\mathrm{ent}}}{\partial p\_{i j}}=-\log p\_{i j}-1$.
> By applying the chain rule, we obtain the gradient of the entropy loss with respect to the logits as follows:
> $\frac{\partial \mathcal{L}\_{\text {ent}}}{\partial z\_{i k}}=\sum\_{j=1}^K \frac{\partial \mathcal{L}\_{\text {ent}}}{\partial p\_{i j}} \cdot \frac{\partial p\_{i j}}{\partial z\_{i k}}=\sum_{j=1}^K\left(-\log p\_{i j}-1\right) \cdot p\_{i j}\left(\delta\_{j k}-p\_{i k}\right)$, where the Kronecker delta: $\delta_{j k} = \{ 1 \text{ if } j = k, \text{ else } 0 \}$.
>
> This expression shows that entropy minimization encourages the model to produce increasingly (over) confident (i.e., low-entropy) predictions by amplifying the largest logit and suppressing the others, which is also discussed in [10].
> To address prediction bias in multi-modal domain shifts, PDR partitions each data batch into biased and unbiased subsets using a prediction bias indicator (Eq. (1)), which leverages batch-average prediction frequency to identify overconfident predictions, often false positives due to entropy minimization. These biased samples are regularized via entropy maximization (the second term in Eq. (2)), which counteracts noise propagation by encouraging uniform probability distributions, as supported by theoretical insights from [a, b, 31] on entropy regularization under noisy conditions.
>
> Additionally, to balance bias and confidence levels and prevent issues like unreliable gradient from low-confidence predictions [19], we propose a quantile ranking strategy (Alg. 1). This strategy reweights unbiased samples by assigning higher weights to confident predictions and lower weights to uncertain ones, ensuring stable and reliable test-time adaptation. The theoretical reasonability of this approach stems from its alignment with robust optimization principles, where reweighting mitigates the impact of noisy outliers, as discussed in [19]. We demonstrate PDR’s consistent stability and superior performance across challenging multi-modal domain shift scenarios, including continual settings (Table 8, Appendix D) and changing environments (Section 4.4).
>
> We  will add the above theoretical analysis and discussion in the Appendix.
>
>
> [a] J. Liang, D. Hu, Y. Wang, R. He and J. Feng, "Source Data-Absent Unsupervised Domain Adaptation Through Hypothesis Transfer and Labeling Transfer," in IEEE Transactions on Pattern Analysis and Machine Intelligence, vol. 44, no. 11, pp. 8602-8617, 1 Nov. 2022, doi: 10.1109/TPAMI.2021.3103390.
>
> [b] J. Li, Z. Yu, Z. Du, L. Zhu and H. T. Shen, "A Comprehensive Survey on Source-Free Domain Adaptation," in IEEE Transactions on Pattern Analysis and Machine Intelligence, vol. 46, no. 8, pp. 5743-5762, Aug. 2024, doi: 10.1109/TPAMI.2024.3370978.
>
> $\textbf{W2}$: The meaning of $Z$ and $\mathcal{K}$.
>
> $\textbf{A2}$: $Z$ represents the frequency of predicted class for $x$, which is a scalar. $\bar{Z}$ is also a scalar since it denotes the average frequency of the predicted labels within a batch. $\mathcal{Z}$ is a vector, consting of all instances' bias level computed by Eq. (1). And $\mathcal{K}$ is also a vector, representing the softmax confidence. We will clarify the meaning of $Z$ and $\mathcal{K}$, and bold the vectors to improve the presentation for better clarity, following your advice.

---

> > ### Comment · Reviewer_h6TZ · 2025-08-05
> >
> > The response has adequately addressed my primary concerns.
> >
> > I encourage the authors to include the full theoretical analysis in a future revision. With the necessary revisions, I believe this work has strong potential to broaden the applications of test-time adaptation research.

---

> > > ### Author Response · Authors · 2025-08-05
> > >
> > > We truly appreciate your recognition of our work’s potential to broaden the scope of test-time adaptation research. We will include the full theoretical analysis in a future revision, following your advice.

---

### Comment · Area_Chair_RbgD · 2025-08-04
**Reminder: Review Rebuttal and Submit Final Justification**

Dear Reviewers,

As we approach the end of the author–reviewer discussion phase (**Aug 6, 11:59pm AoE**), I kindly remind you to read the author rebuttal carefully, especially any parts that address your specific comments. Please consider whether the response resolves your concerns, and if not, feel free to engage in further discussion with the authors while the window is still open.

Your timely participation is important to ensure a fair and constructive review process. If you feel your concerns have been sufficiently addressed, you may also submit your Final Justification and update your rating early. Thank you for your contributions.

Best,

ACs

---

### Note · Authors · 2025-08-12

We sincerely thank all reviewers and AC for their valuable time and efforts. We are encouraged that reviewers appreciate the novelty (Reviewer#h6TZ), significance (Reviewer#h6TZ, Reviewer#Y3nk, Reviewer#UQ6f), presentation quality (Reviewer#xcN3, Reviewer#Jc1F, Reviewer#UQ6f), and effectiveness (Reviewer#h6TZ, Reviewer#xcN3, Reviewer#Y3nk, Reviewer#UQ6f) of our paper.

We provide earnest rebuttals for all reviewers:
- For Reviewer#h6TZ,  we provide a theoretical analysis to clarify the reasonability of our method’s design, and we make precise definitions of some notations.
- For Reviewer#xcN3, we add additional experiments to demonstrate the effectiveness of our method and offer experimental details for transparency.
- For Reviewer#Y3nk, we offer more supporting evidence to highlight our motivation.
- For Reviewer#Jc1F, we provide deeper analysis to hopefully inspire future research. And we also present additional experiments to demonstrate the broader applicability of our method.
- For Reviewer#UQ6f, we clarify some notations and equations for better understanding.

We are pleased that the reviewers confirmed their main concerns have been adequately addressed. To further improve the paper, we will integrate all relevant discussions and empirical results into the final version.

Finally, we thank the reviewers for acknowledging our work’s potential to broaden the application of test-time adaptation research. And we extend our gratitude to all the reviewers and AC for your valuable feedback, which has improved our paper to a better version.

---

### Decision · Program_Chairs · 2025-09-17

**Decision:**

Accept (spotlight)

**Comment:**

All reviewers support the acceptance of this paper. After a careful reading of the manuscript, the detailed rebuttal, and all reviewer comments, I agree with their overall assessment. The paper addresses an important and under-explored problem, multi-modal test-time adaptation under simultaneous domain shifts, and proposes a novel and well-motivated method (PTA) that is both empirically effective and theoretically grounded. The authors provide extensive experiments, thorough ablations, and insightful rebuttals that satisfactorily resolve most reviewer concerns. I encourage the authors to further refine the camera-ready version by incorporating the reviewers’ constructive suggestions, especially regarding clarity, reproducibility, and additional implementation details. Overall, this is a solid contribution to the area of test-time adaptation and multi-modal learning.